# Maternal balanced energy-protein supplementation reshapes the maternal gut microbiome and enhances carbohydrate metabolism in infants: a randomized controlled trial

Balanced energy-protein (BEP) supplementation during pregnancy and lactation can improve birth outcomes and infant growth, with the gut microbiome as a potential mediator. The MISAME-III randomized controlled trial (ClinicalTrial.gov: NCT03533712) assessed the effect of BEP supplementation, provided during pregnancy and the first six months of lactation, on small-for-gestational age prevalence and length-for-age Z-scores at six months in rural Burkina Faso. Nested within MISAME-III, this sub-study examines the impact of BEP supplementation on maternal and infant gut microbiomes and their mediating role in birth outcomes and infant growth. A total of 152 mother-infant dyads ($n = 71$ intervention, $n = 81$ control) were included for metagenomic sequencing, with stool samples collected at the second and third trimesters, and at 1–2 and 5–6 months postpartum. BEP supplementation significantly altered maternal gut microbiome diversity, composition, and function, particularly those with immune-modulatory properties. Pathways linked to lipopolysaccharide biosynthesis were depleted and the species *Bacteroides fragilis* was enriched in BEP-supplemented mothers. Maternal BEP supplementation also accelerated infant microbiome changes and enhanced carbohydrate metabolism. Causal mediation analyses identified specific taxa mediating the effect of BEP on birth outcomes and infant growth. These findings suggest that maternal supplementation modulates gut microbiome composition and influences early-life development in resource-limited settings.

In 2022, an estimated 148.1 million children under the age of five—approximately 22% of this age group globally—are affected by stunted growth, which limits their full developmental potential[1]. Stunting commonly begins before birth, known as fetal growth restriction, and accumulates during infancy until the age of 18 to 23 months; after two years of age, the growth impairment tends to be largely irreversible[2–5]. In low- and middle-income countries, maternal undernutrition is the main cause of low birth weight and small-for-gestational age (SGA) in infants, and poor linear growth in children[2,5,6]. While studies have shown that providing lipid-based nutrient supplements to pregnant

✉ e-mail: carl.lachat@ugent.be; Trenton@Dailey-Chwalibog.com

women can positively affect birth weight, length, and SGA[7], evidence of the impact of maternal supplementation on infant growth remains limited and inconsistent to date[8–12]. This suggests the need to explore underlying biological pathways that may mediate these effects.

One such potential pathway is the infant gut microbiota, which undergoes rapid maturation during early childhood[13]. Healthy children experience a patterned ecological assembly of their gut microbiome during the first two years of life[14], a process influenced by geography[15–17], mode of delivery[18], feeding practices[18,19], and health status[20]. A mature gut microbiota during the first month of life may influence infant growth trajectories during the first year of life[21]. Inadequate maturation of the gut microbiota might contribute to the onset of child undernutrition, and both microbiota immaturity and malnutrition can be partially ameliorated following nutrition interventions[22,23]. Importantly, the plasticity of the gut microbiome presents a promising avenue for nutritional interventions aimed at improving early-life growth and development.

However, research on how maternal nutritional supplementation influences the infant gut microbiome remains sparse. Balanced energy–protein (BEP) supplements, which are fortified with multiple micronutrients and provide energy and protein in a balanced composition (<25% of total energy from protein), have been shown to reduce the risk of stillbirth and SGA[24]. This analysis is part of a biospecimen sub-study (BioSpé)[25] nested within the MIcronutriments pour la SAnté de la Mère et de l'Enfant-III (MISAME-III) randomized controlled trial, conducted across six rural health centers in Burkina Faso. In the intervention group, women received BEP supplementation during pregnancy and the first six months of lactation, in addition to iron folic acid (IFA) tablets during pregnancy and the first six weeks postpartum. The control group received IFA tablets only.

A prior study has shown that the diet of women who participated in the MISAME-III trial was nondiverse, with only 45% of participants reaching Minimum Dietary Diversity for Women (i.e., ≥5 food groups)[26]. The base diet is mainly cereal-based, with leafy vegetables as a supplement, contributing to half the total energy intake; other nutritious food groups, such as fruit, dairy, eggs, fish, and meat contribute only a small proportion to the total energy intake[27,28]. BEP supplementation significantly improved both energy intake and micronutrient adequacy without displacing other nutrient sources. On average, women in the intervention group consumed an additional 15.2 g of protein, 37.5 g of carbohydrates, and 25.6 g of fat compared to the control group on a daily basis[28]. This supplement led to modest improvements in birth weight and length and reduced the prevalence of low birth weight[29]. Benefits on size at birth persisted, as reflected in improved linear growth (i.e., length-for-age Z-scores, LAZ) at six months of age[9].

Here, we hypothesize that the individual gut microbiome may mediate the effects of BEP supplementation on infant anthropometric outcomes. Using deep metagenomic sequencing, we evaluate changes in gut microbiota composition and function in both mothers and their infants due to maternal BEP supplementation, with a particular focus on microbial metabolic pathways that have not been explored in previous studies. We further explore whether maternal gut microbiome mediates the effects of prenatal BEP supplementation on birth outcomes and whether the infant gut microbiome induces the impact of maternal BEP supplementation on infant growth.

## Results
### Study population
Participants for this sub-study were selected based on gestational age criteria to prioritize second-trimester recruitment, ensuring a minimum sample size of 60 across most time points. Accounting for a 20% drop-out rate, we aimed to select 75 participants per group, balancing feasibility with achieving sufficient statistical power for longitudinal analysis. A total of 152 mother-infant pairs were included, with 71 pairs in the intervention group and 81 in the control group. We collected 731 fecal samples, comprising 256 infant samples collected at 1–2 months and 5–6 months of life, and 475 maternal samples collected during the second trimester (i.e., inclusion), third trimester, 1–2 months and 5–6 months postpartum (Fig. 1). Sample collection during the second trimester was more limited, with roughly 20 participants from each group providing samples, whereas at later time points, approximately 70 pairs from each group provided samples. Overall, apart from the second trimester, 105 mother-infant dyads ($n = 55$ in the intervention group and $n = 50$ in the control group) provided samples at all time points.

The baseline characteristics of the subset population were comparable to the full MISAME-III cohort, except for a slightly lower percentage of anemia in the intervention group within the selected subset, which was not statistically significant (Table S1). The mean (standard deviation, SD) age of the subset participants (pregnant women) was 24.3 (5.55) years, and the body mass index (BMI) was 22.4 (3.5) kg/m² at enrollment. Over half of the women (58.5%) had no formal education, 69.2% came from food-insecure households, 59.3% had no formal employment, 38.4% had at least three prior pregnancies, and 30.5% were anemic. Nearly all infants were exclusively breastfed, with an average breastfeeding duration of 5.78 months, with 5.91 months in the intervention group and 5.67 months in the control group. Among BEP-supplemented participants, the average compliance rate was 83.1% during the prenatal and 86.0% during the postnatal supplementation periods. Overall, maternal BMI, gestational age, and household wealth index did not differ significantly between groups (Fig. 1, Table S1).

The mean (SD) gestational age at birth was 39.8 (1.72) weeks, and the gestational age at which the intervention began was 9.85 (3.58) weeks, with no significant differences observed between the intervention and control groups. The mean (SD) infant weight and length at birth were 3.0 (0.47) kg and 48.5 (2.16) cm, respectively, and were comparable between the intervention and control groups. While infants in the intervention group exhibited modestly higher weights during the first six months of life compared to those in the control group, these differences were not statistically significant. This difference was also reflected in higher Weight-for-Age Z-scores (WAZ) and Weight-for-Length Z-scores (WLZ) during the first six months of life in the intervention group. However, these differences were not statistically significant. In contrast, LAZ remained relatively stable and comparable between the two groups throughout the study period (Fig. 1).

### Deep sequencing of the gut microbiome and early colonization
The average (SD) total sequencing depth of infant and maternal metagenomes was 8.03 (1.87) and 25.29 (5.28) gigabase-pairs, respectively (Fig. S1). On average, 7.4% (1.7%) of maternal and 19.9% (18.8%) of infant metagenomic reads were removed during quality control and human read decontamination processes. After filtering, 58.1% (14.2%) of the infant reads and 58.0% (3.0%) of the maternal reads could be robustly assigned to genomes in our microbial genome database. The maternal gut microbiomes were dominated by *Prevotella* species, which accounted for a median of 37.9% of the microbial composition, with no major shift between pre- and postnatal compositions (38.1% vs 37.7%). In contrast, the infant gut microbiomes were more variable, with the abundance of *Prevotella* ranging from 0.0% to 83.9% of the microbial composition, present in 44 out of the 256 infant samples. In most infant samples, *Bifidobacterium* strains were dominant, comprising a median of 44.5% of the microbial compositions and present in 98.0% of infant samples (Fig. 2A).

### BEP supplementation affects the maternal gut microbiome diversity and gene richness
The microbial richness of the infant gut was markedly lower than that of the maternal gut, with an observed species index ranging from 2 to 43 in infants and from 75 to 614 in mothers (Fig. 2B). This difference was further emphasized by the Shannon diversity index, which had a

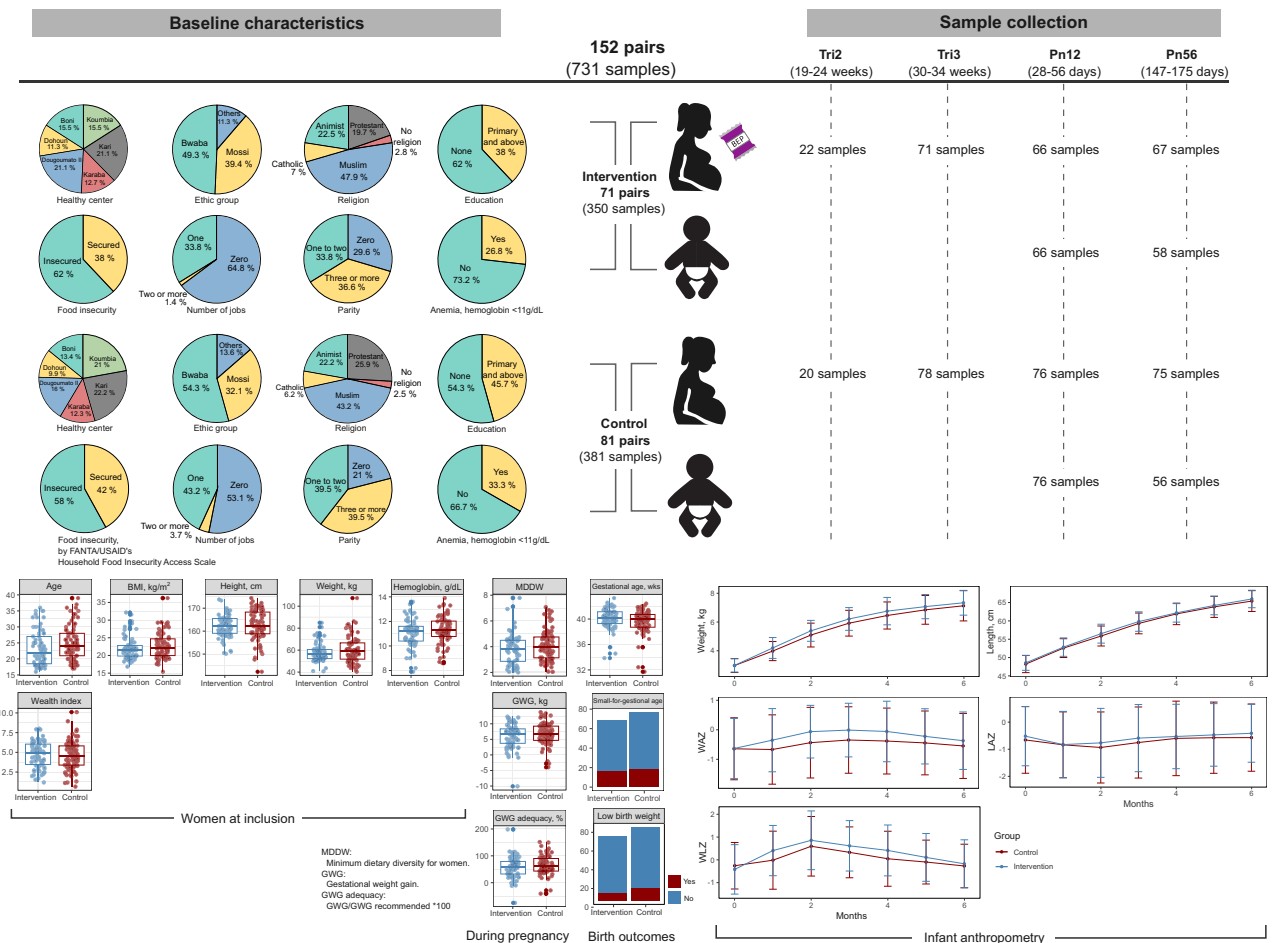

**Fig. 1 | Baseline characteristics, maternal and infant anthropometry, and sample collection timeline.** Pie charts depict the baseline characteristics of the study participants, including health center, ethnicity, religion, education level, food insecurity, number of jobs, parity, and anemia for each group. Box plots present other measures, such as the age of the participants, BMI, height, weight, hemoglobin levels, and wealth index, along with minimum dietary diversity score and gestational weight gain during pregnancy. Birth outcomes are displayed using box plots and bar plots. Infant anthropometry data are presented as growth curves for weight, length, and corresponding Z-scores (WAZ, LAZ, and WLZ) during the first six months of life, comparing the intervention and control groups. Box plots indicate median (middle line), 25th, 75th percentile (box), and the smallest and largest values within 1.5 × interquartile range (IQR) from the box (whiskers). Data points beyond this range are considered outliers (single points). Data in line plots are presented as mean ± standard error. BMI body mass index, GWG gestational weight gain, MDDW minimum dietary diversity for women, Pn12 postnatal 1–2 months, Pn56 postnatal 5–6 months, Tri2 trimester 2, Tri3 trimester 3, LAZ length-for-age Z-score, WAZ weight-for-age Z-score, WLZ weight-for-length Z-score.

median of 1.48 bits in infant samples compared to 4.47 bits in maternal samples. Shannon diversity was higher in prenatal samples compared to postnatal samples, but the difference was not statistically significant ($p = 0.08$, Wilcoxon rank sum test), with median values of 4.53 bits and 4.10 bits, respectively (Fig. 2C).

Using the Targeted Maximum Likelihood Estimator (TMLE), no significant average treatment effect of BEP supplementation on Shannon diversity was estimated when analyzing time points separately. Due to the limited number of samples from the second trimester and the similarity in Shannon diversity between the second and third trimester, these time points were combined for further analysis. In this combined analysis, BEP supplementation was estimated to increase maternal prenatal Shannon diversity index by 0.144 bits (95% CI: 0.009–0.280; $p = 0.037$), which corresponds to an estimated increase of 1.16 equally-common species.

As illustrated in Fig. 2D, beta-diversity analyses showed diversification of the microbial communities, with significant differences between samples taken at different stages of pregnancy ($p = 0.002$) or infancy ($p = 0.001$). However, no significant inter-community difference was attributed to BEP supplementation in either maternal ($p = 0.098$) or infant samples ($p = 0.960$).

Similarly, microbial gene richness was lower in infant samples compared to maternal samples, with a median observed gene richness of 27,558 in infants and 36,340 in mothers (Fig. 2E). TMLE analysis suggested that BEP supplementation increased gene counts in maternal samples at 1–2 months postpartum by 2007 counts (95% CI: 95–3920; $p = 0.040$). No significant associations between BEP supplementation and gene counts were found at the other time points for maternal or infant samples. Although beta-diversity of gene data showed diversification of the infant metagenome over time ($p = 0.003$), no significant differences were attributed to the intervention ($p = 0.155$). Similarly, in infant samples, the metagenome showed significant diversification over time ($p = 0.002$), but no significant differences could be attributed to the intervention ($p = 0.704$) (Fig. 2F). These results indicate that temporal changes in the gut microbiome play a more prominent role in shaping microbial gene diversity than the intervention itself.

## BEP supplementation is associated with microbial temporal stability

In maternal samples, the majority of species were consistently present across all time points in both the intervention and control groups.

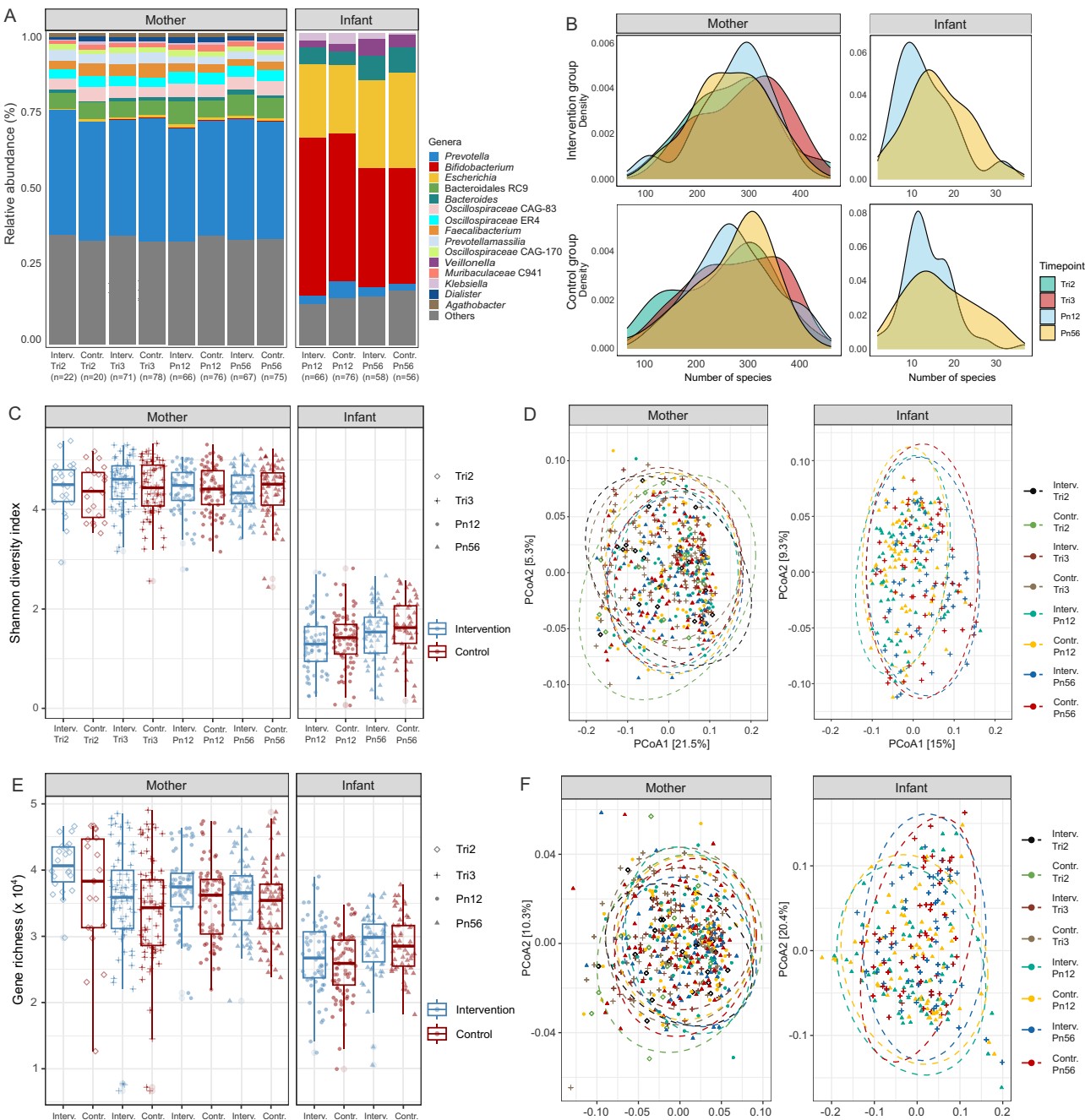

**Fig. 2 | Relative abundance, diversity, and gene richness in maternal and infant samples across time points by groups. A** Relative abundance of the top 15 most abundant genera (averaged across all samples) aggregated by treatment groups and sample collection time points. Genera with lower abundance are grouped into the "Other" category. **B** Density plots showing the total number of microbial species detected per sample in maternal and infant stool samples. Box-and-jitter plots of (**C**) Shannon diversity indices and (**E**) gene richness grouped by intervention and control at each time point. Maternal samples: Interv.Tri2 ($n = 22$); Contr.Tri2 ($n = 20$); Interv.Tri3 ($n = 71$); Contr.Tri3 ($n = 78$); Interv.Pn12 ($n = 66$); Contr.Pn12 ($n = 76$); Interv.Pn56 ($n = 67$); Contr.Pn56 ($n = 75$). Infant samples: Interv.Pn12 ($n = 66$); Contr.Pn12 ($n = 76$); Interv.Pn56 ($n = 58$); Contr.Pn56 ($n = 56$). Box plots indicate median (middle line), 25th, 75th percentile (box), and the smallest and largest values within 1.5 × interquartile range (IQR) from the box (whiskers). Data points beyond this range are considered outliers (single points). Principal Coordinate Analysis (PCoA) ordination plots of (**D**) microbial species based on weighted UniFrac distance and (**F**) gene data based on Bray-Curtis distance. Interv intervention, Contr control, Tri2 trimester 2, Tri3 trimester 3, Pn12 postnatal 1–2 months, Pn56 postnatal 5–6 months.

Firmicutes A, the most abundant phylum, constituted up to 75% of the microbial community at all time points. In contrast, infant samples exhibited less strains recurrence, with fewer than half of the strains consistently present across all time points (Fig. 3A). Across all samples, 2,654 unique microbial genomes were detected in at least one sample, of which 2,493 were exclusive to the maternal population, 396 were unique to the infant population, and 235 were shared between mothers and infants. Among the shared microbial genomes, 205 were already observed in the mothers during prenatal time points. Approximately 50% of the microbial genomes found in the infant gut were also detected in at least one maternal sample, and this fraction remained relatively stable over time, with 36% in the second trimester, 45% in the third trimester, 44% at 1–2 months postpartum, and 40% at 5–6 months postpartum.

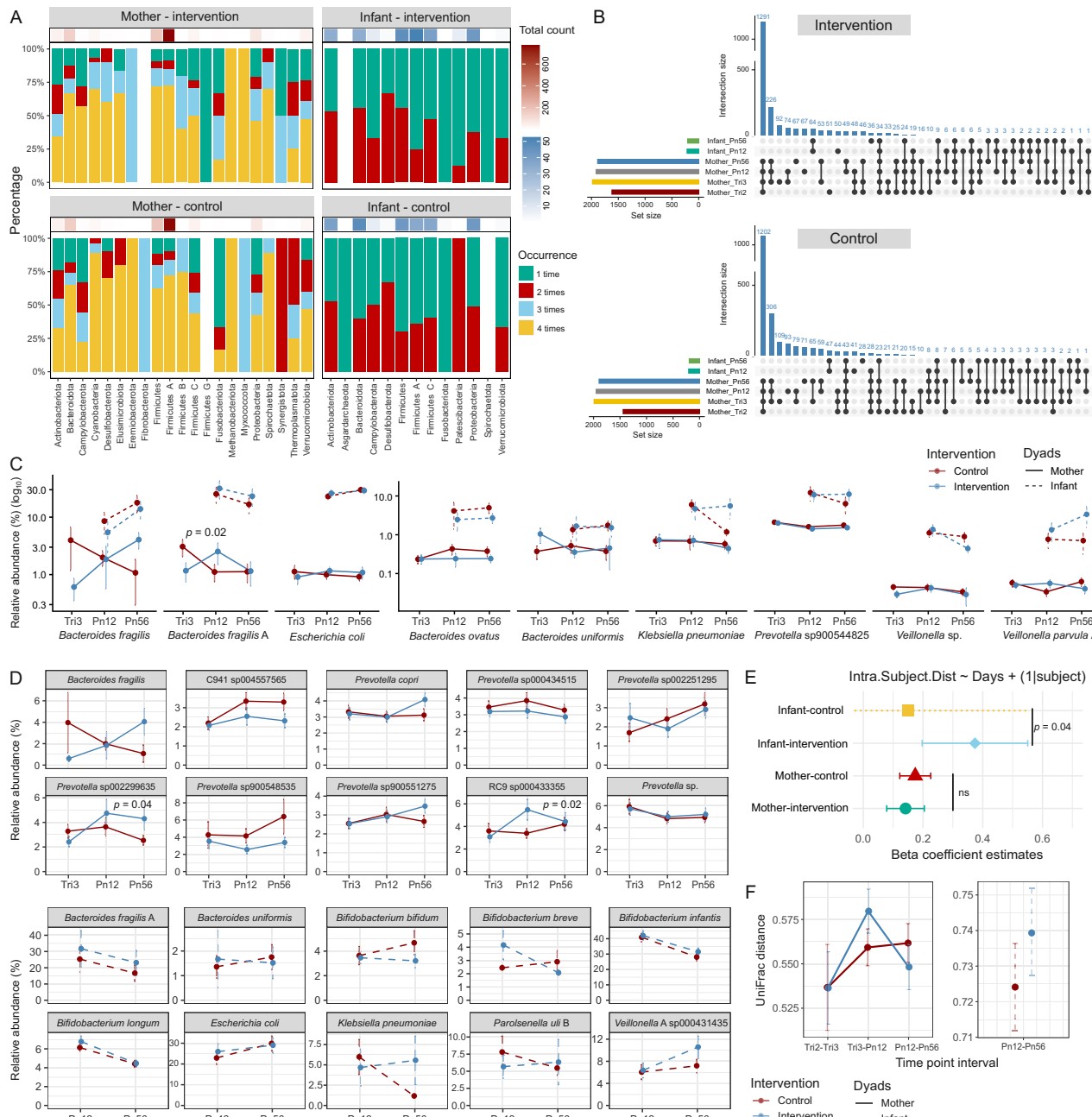

**Fig. 3 | Temporal stability of maternal and infant gut microbiomes across time points by groups. A** Bar plots of taxa recurrence rates within each phylum. **B** UpSet plots showing the number of shared strains across different time points and mother-infant pairs. Line plots displaying the relative abundance of (**C**) taxa shared between mothers and infants across time points and (**D**) the top ten most abundant species in both mother and infant samples. The species shown have a minimum abundance of 1% and are present in at least five samples across the time points (excluding the second trimester due to limited measurements). Data are presented as mean ± standard error. **E** Linear regression between dissimilarity (beta diversity) against collection date interval, comparing intra-subject microbiome stability across groups. Data are presented as the estimated effect size of the predictor (days interval) on the outcome (intra subject distance) obtained from the mixed-effects model, and the lower and upper bounds of the 95% confidence interval on the effect estimate. The Infant-control group has infinite lower and upper confidence intervals (dashed line). Number of subjects: Infant-control $n = 52$; Infant-intervention $n = 54$; Mother-control $n = 79$; Mother-intervention $n = 68$. **F** Shifts in microbiome over time, assessed by intra-subject distance using weighted UniFrac. Data are presented as mean ± standard error. Number of subjects: Infant-control $n = 52$; Infant-intervention $n = 54$; Mother-control $n = 79$; Mother-intervention $n = 68$. Tri2 trimester 2, Tri3 trimester 3, Pn12 postnatal 1–2 months, Pn56 postnatal 5–6 months.

Interestingly, a higher number of shared genomes was observed between mothers and infants in the control group compared to the intervention group ($p = 0.019$, chi-square test). Specifically, 165 out of 281 genomes were shared in the control group, whereas 142 out of 292 were shared in the intervention group (Fig. 3B). Of the nine shared strains present in at least five samples at each time point, the relative abundances of these shared strains were notably lower in mothers compared with their infants (Fig. 3C). Additionally, a strain of *Bacteroides fragilis* exhibited significantly greater increases in relative abundance in the intervention group compared to the control group from the third trimester to 1–2 months postpartum ($p = 0.02$).

Figure 3D presents the top ten taxa with a minimum abundance of 1% and presence in at least five samples across all time points, categorized by intervention groups, for both maternal and infant samples.

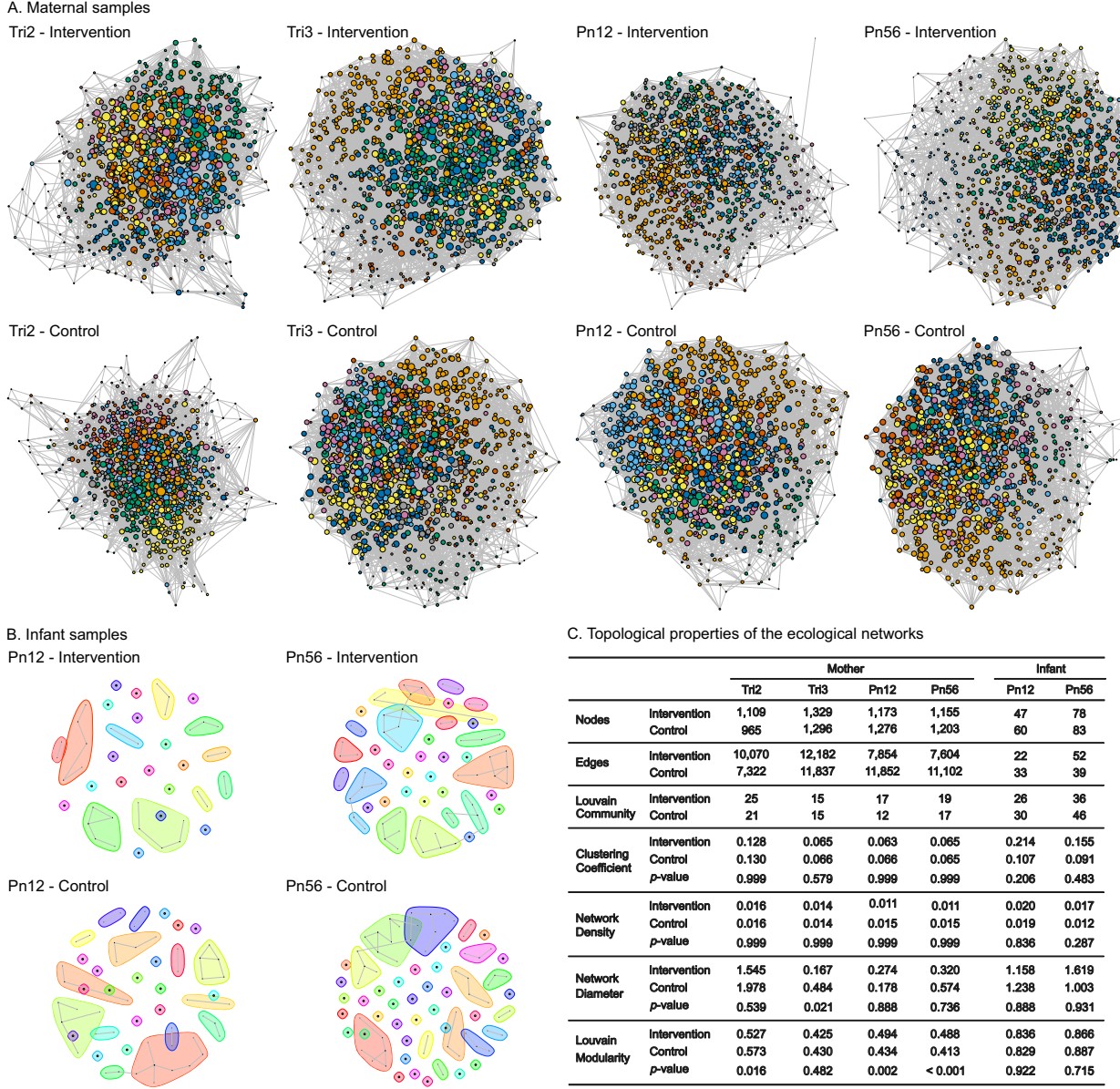

**Fig. 4 | Co-occurrence network analysis of maternal and infant microbiomes across time points by groups.** Co-occurrence networks for (**A**) maternal samples at four time points and (**B**) infant samples at two time points. Nodes represent microbial taxa, with edges indicating significant co-occurrence relationships between taxa. Node size is proportional to the degree (number of connections), and colors indicate different communities identified by Louvain clustering. **C** Topological properties of the ecological networks depicting co-occurrence of the gut microbiome, including nodes, edges, clustering coefficient, network density, network diameter, Louvain community count, and modularity for both maternal and infant samples. Significant differences between the intervention and control groups were determined using a two-sided permutation test (1000 permutations), with significance set at $p < 0.05$. The $p < 0.001$ indicates a $p = 0$ returned when using 1000 permutations. Tri2 trimester 2, Tri3 trimester 3, Pn12 postnatal 1–2 months, Pn56 postnatal 5–6 months.

A mixed-effects linear regression model, with subject identifier as a random effect, was used to compare changes in relative abundance over time between groups. Among the top ten taxa in maternal samples, two exhibited significantly greater increases in the intervention group compared to the control group: *Prevotella* sp002299635 (from the third trimester to 5–6 months postpartum; $p = 0.04$) and Bacteroidales RC9 sp000433355 (from the third trimester to 1–2 months postpartum; $p = 0.02$).

Our analysis revealed that the infant gut microbiome in the intervention group changed more rapidly over time compared to the control group ($p = 0.04$), as evidenced by mixed-effects linear regression models with intra-subject distance (calculated using weighted UniFrac) as the dependent variable and days interval between consecutive time points as the independent variable (Fig. 3E). Similarly,

Fig. 3F revealed distinct temporal patterns in microbiome composition. In maternal samples, microbiome dissimilarity increased progressively from the second to the third trimester, peaked between late pregnancy and early postpartum, and then stabilized. The shift was more pronounced in the intervention group, but it was not statistically significant. In infants, microbiome composition exhibited greater shifts over time compared to maternal samples, with slightly higher UniFrac distances observed in the intervention group, though this difference was not statistically significant.

## BEP supplementation alters the network structure of the maternal gut microbiome over time

The co-occurrence network analysis revealed distinct shifts in microbial community structure between the intervention and control

groups in maternal samples, particularly in relation to network modularity (Fig. 4). During prenatal time points, the intervention group exhibited a higher number of nodes and edges compared to the control group. In the second trimester, Louvain community detection identified more communities in the intervention group, alongside significantly lower modularity ($p = 0.016$), suggesting less compartmentalization and more interconnectivity within the microbial network. In the third trimester, although network density and clustering coefficients were similar between groups, the network diameter in the intervention group was significantly smaller ($p = 0.021$), reflecting more compact interactions between taxa. During postnatal time points, the network structure shifted, with the intervention group displaying fewer nodes and edges but significantly greater modularity at both 1–2 months postpartum ($p = 0.002$) and 5–6 months postpartum ($p < 0.001$), suggesting that BEP supplementation led to a more modular and compartmentalized network over time.

## BEP supplementation is associated with maternal microbial abundances

Differential abundance analyses were conducted separately at each time point, as well as across time points (excluding the second trimester due to limited measurements), with the individual subject included as a random effect to account for repeated measures. In the individual time point analyses (Fig. 5A), 29 taxa in maternal samples were significantly associated with the intervention, 16 of which had a $\log_2$ fold change ($\log_2$FC) greater than 0.5. Among these, two species exhibited larger biological effect sizes: depletion of an *Oscillospiraceae* species (CAG-103 sp000432375; $p = 2.48 \times 10^{-4}$; $\log_2$FC = −1.12) in the second trimester and enrichment of *B. fragilis* ($p = 3.26 \times 10^{-8}$; $\log_2$FC = 1.31) at 5–6 months postpartum. In contrast, no robustly significant associations were found between BEP supplementation and the microbial abundances in infant stool samples at either time point. Analogously, in the combined time point analysis, no significant associations were found between the intervention and microbial abundance in either maternal or infant samples (Fig. 5B). Figure 5C displays the relative abundance of the 16 taxa that were significantly different between the intervention and control groups. The full result table of differential abundance analyses is available in Supplementary Data 1.

## BEP supplementation alters microbial pathways in mothers and infants

Gene set enrichment analysis indicated concerted enrichment and depletion of certain microbial pathways in response to BEP supplementation at each time point (Fig. 6). However, no significant association was observed between the intervention and the abundance of individual microbial genes.

One notable finding was the depletion of the gene set for the phosphotransferase system (PTS) in stool samples from BEP-supplemented mothers, both pre- and postpartum [$p = 2.56 \times 10^{-5}$ at third trimester (Tri3); $p = 2.23 \times 10^{-3}$ at 5–6 months postpartum (Pn56)]. Conversely, this pathway was enriched in stool samples from infants born to BEP-supplemented mothers [$p = 2.67 \times 10^{-2}$ at 1–2 months postpartum (Pn12); $p = 2.76 \times 10^{-3}$ at Pn56]. Several other pathways showed significant depletion in BEP-supplemented mothers at multiple time points, including lipopolysaccharide (LPS) biosynthesis ($p = 1.53 \times 10^{-4}$ at Tri3; $p = 1.88 \times 10^{-2}$ at Pn56), ABC transporters ($p = 2.27 \times 10^{-8}$ at Tri3; $p = 1.17 \times 10^{-3}$ at Pn56), biosynthesis of cofactors ($p = 2.27 \times 10^{-8}$ at Tri3; $p = 1.42 \times 10^{-2}$ at Pn56), and porphyrin metabolism ($p = 2.85 \times 10^{-3}$ at Tri3; $p = 1.74 \times 10^{-2}$ at Pn56). In contrast, the ribosome complex pathway was significantly enriched in BEP-supplemented mothers at multiple time points ($p = 1.55 \times 10^{-6}$ at Tri3; $p = 1.17 \times 10^{-3}$ at Pn56). Other pathways, such as flagellar assembly, showed a complex dynamic pattern, being enriched at inclusion (Tri2, $p = 9.35 \times 10^{-5}$) but depleted at later time points ($p = 1.45 \times 10^{-5}$ at Tri3

and $p = 2.95 \times 10^{-8}$ at Pn12) in BEP-supplemented mothers. Interestingly, the gene set for pathogenic *Escherichia coli* infection was depleted in the second trimester in BEP-supplemented mothers (Supplementary Data 2). The full table of significantly differential pathways is available in Supplementary Data 2.

Figure S2A depicts the gene set depletion of the LPS biosynthesis pathway, illustrating that specific sections of sequential genes in the pathway were affected during the third trimester. All genes involved in the conversion from UDP-N-acetyl-D-glucosamine to Lauroyl-KDO2-lipid IV(A) exhibited reduced abundance, with $\log_2$FC ranging from −0.014 (*LpxH*) to −0.136 (*LpxI*). Figure S2B shows that, of the six genes in the PTS enriched in infants at 5–6 months of age born to BEP-supplemented mothers, half were specific to the cellobiose subunit. Analogous pathway visualizations for PTS, flagellar assembly, and ABC transporters at the third trimester are available in the Supplementary Information (Fig. S3-S5).

## Microbiome modulation mediates birth outcomes and infant growth in response to BEP supplementation

We conducted causal mediation analyses to elucidate the role of the microbiome (through changes in either Shannon diversity or individual taxon abundance) in the downstream effects of BEP supplementation on birth outcomes (i.e., SGA prevalence, gestational age, birth weight and length) and infant growth metrics (i.e., stunting prevalence, LAZ, WAZ and WLZ) at 3 and 6 months of age.

The mediation analyses using the Shannon diversity of the microbial communities as a mediator revealed significant natural direct effects of the intervention on gestational age of the infants at birth (0.56 weeks; 95% CI: 0.05–1.08 weeks), as well as on WAZ at 3 months (0.43; 95% CI: 0.03–0.83) and at 6 months (0.43; 95% CI: 0.04–0.82), and the odds of stunting at 6 months (−11.6%; 95% CI: −21.5– −1.6%) (Fig. 7A). However, no significant natural indirect effects were observed, indicating that these effects were not mediated by shifts in microbial diversity. Therefore, we conducted a multivariate mediation analysis using individual taxon abundances.

In the second trimester, specific taxa exhibited a significant mediating effect on birth outcomes. The abundances observed after the intervention of species from order Oscillospirales (UMGS911 sp900545935), *Lachnospira* (sp900316325), *Faecalibacterium prausnitzii*, and *Butyricicoccus* A (sp002395695) had significantly positive effects on gestational age, birth weight, and length. Conversely, the intervention-altered abundances of the order Bacteroidales (RC9 sp900546925 and RC9 sp000434935), *Oscillospiraceae* (CAG-83 sp900547745), *Phascolarctobacterium succinatutens*, *Prevotella* sp900553155, *Gastranaerophilaceae* (UMGS1585), and *Prevotellamassilia* sp900543155 had negative effects on anthropometric outcomes (Fig. 7B).

In the third trimester, *Ruminococcus bicirculans* and species from the order Christensenellales (CAG-1435), Bacteroidales (RC9 sp900546925), Oscillospirales (UMGS1002), and *Lachnospiraceae* (CAG-632 sp000431515) were associated with decreases in gestational age, birth weight, and length in mothers receiving BEP supplementation. Meanwhile, *Oscillospiraceae* CAG-110 sp900556935 showed a positive mediating effect on gestational age, birth weight, and length in mothers receiving BEP supplementation, without affecting the odds of SGA (Fig. 7C).

At 1–2 months of age, the abundance of *Bifidobacterium breve* showed a significant negative mediating effect on both WAZ (−0.13; 95% CI: −0.24– −0.01) and WLZ (−0.15; 95% CI: −0.28– −0.02) at six months, indicating that higher levels of *B. breve* were associated with lower WAZ and WLZ scores. Conversely, *Streptococcus* sp000187445 exhibited a significant positive mediating effect on WAZ (0.07; 95% CI: 0.01–0.12) and WLZ (0.08; 95% CI: 0.02–0.14), suggesting that its abundance was associated with higher WAZ and WLZ scores (Fig. 7D). *Bifidobacterium infantis* also displayed significant mediating effects on

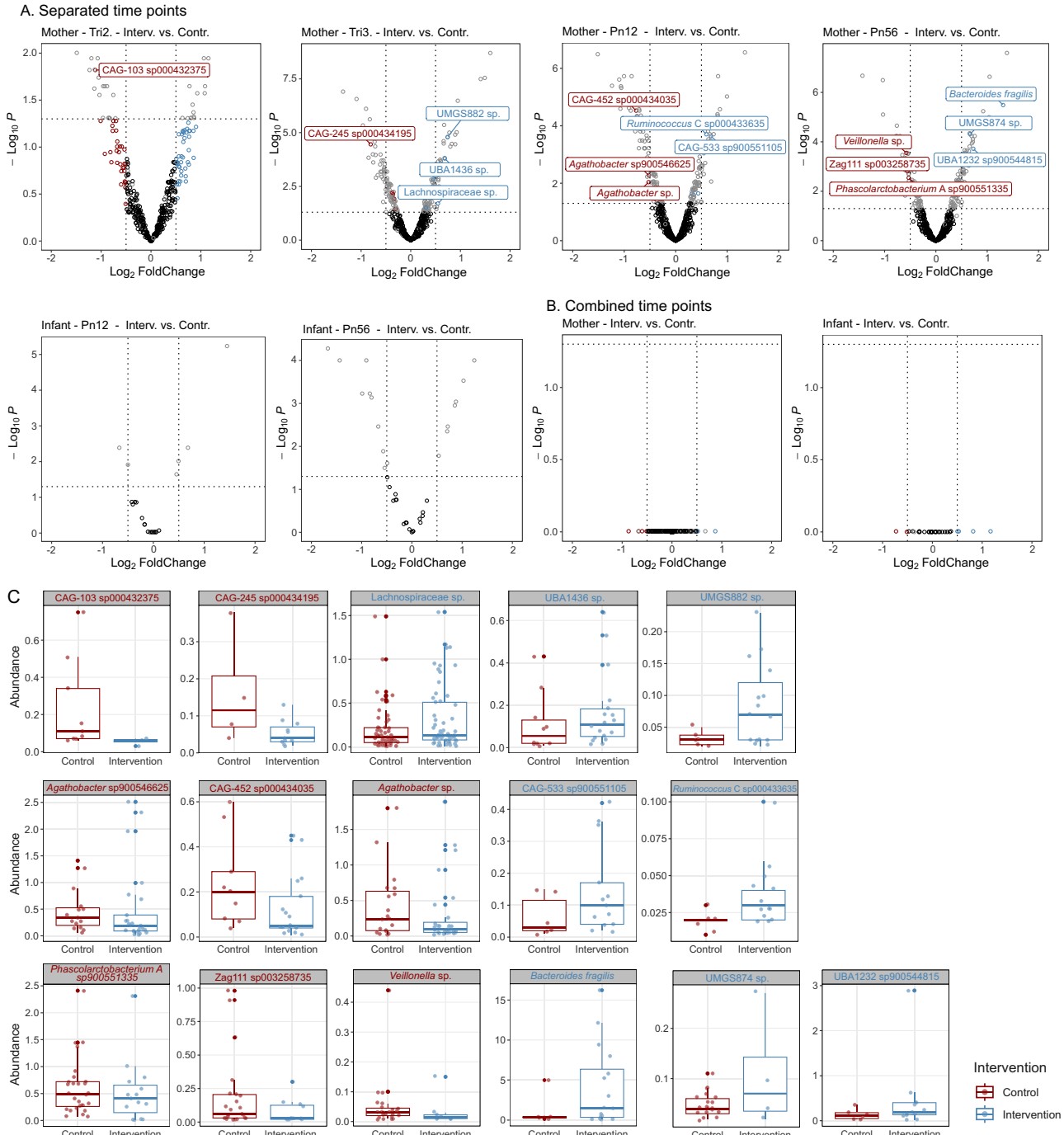

**Fig. 5 | Differential taxa identified by ANCOM-BC2 in maternal and infant samples across time points.** Volcano plots showing differential abundant taxa associated with the intervention (**A**) at each time point for both maternal and infant samples, and (**B**) across all time points combined (excluding the second trimester due to limited measurements). ANCOM-BC2 employs a two-sided test. Each point represents a taxon. The x-axis represents log2 fold change (log$_2$FC) in abundance, and the y-axis represents Benjamini-Hochberg adjusted $p$-values. Horizontal dashed lines indicate the thresholds of statistical significance (i.e., adjusted $p$-values < 0.05), and vertical dashed lines represent |log$_2$FC| > 0.5. Colored dots indicate taxa significantly enriched (blue) and depleted (red) in the intervention group compared to the control group. Results were confirmed through sensitivity analysis for pseudo-counts. Maternal samples: Interv.Tri2 ($n = 22$); Contr.Tri2 ($n = 20$); Interv.Tri3 ($n = 71$); Contr.Tri3 ($n = 78$); Interv.Pn12 ($n = 66$); Contr.Pn12 ($n = 76$); Interv.Pn56 ($n = 67$); Contr.Pn56 ($n = 75$). Infant samples: Interv.Pn12 ($n = 66$); Contr.Pn12 ($n = 76$); Interv.Pn56 ($n = 58$); Contr.Pn56 ($n = 56$). A full table of ANCOM-BC2 outputs can be found in Supplementary Data 1. **C** Boxplots displaying the relative abundance of the 16 taxa with |log$_2$FC| > 0.5 that were significantly different between the intervention and control groups. Only taxa with an abundance > 0 were plotted. Box plots indicate median (middle line), 25th, 75th percentile (box), and the smallest and largest values within 1.5 × interquartile range (IQR) from the box (whiskers). Data points beyond this range are considered outliers (single points). Interv intervention, Contr control, Tri2 trimester 2, Tri3 trimester 3, Pn12 postnatal 1–2 months, Pn56 postnatal 5–6 months.

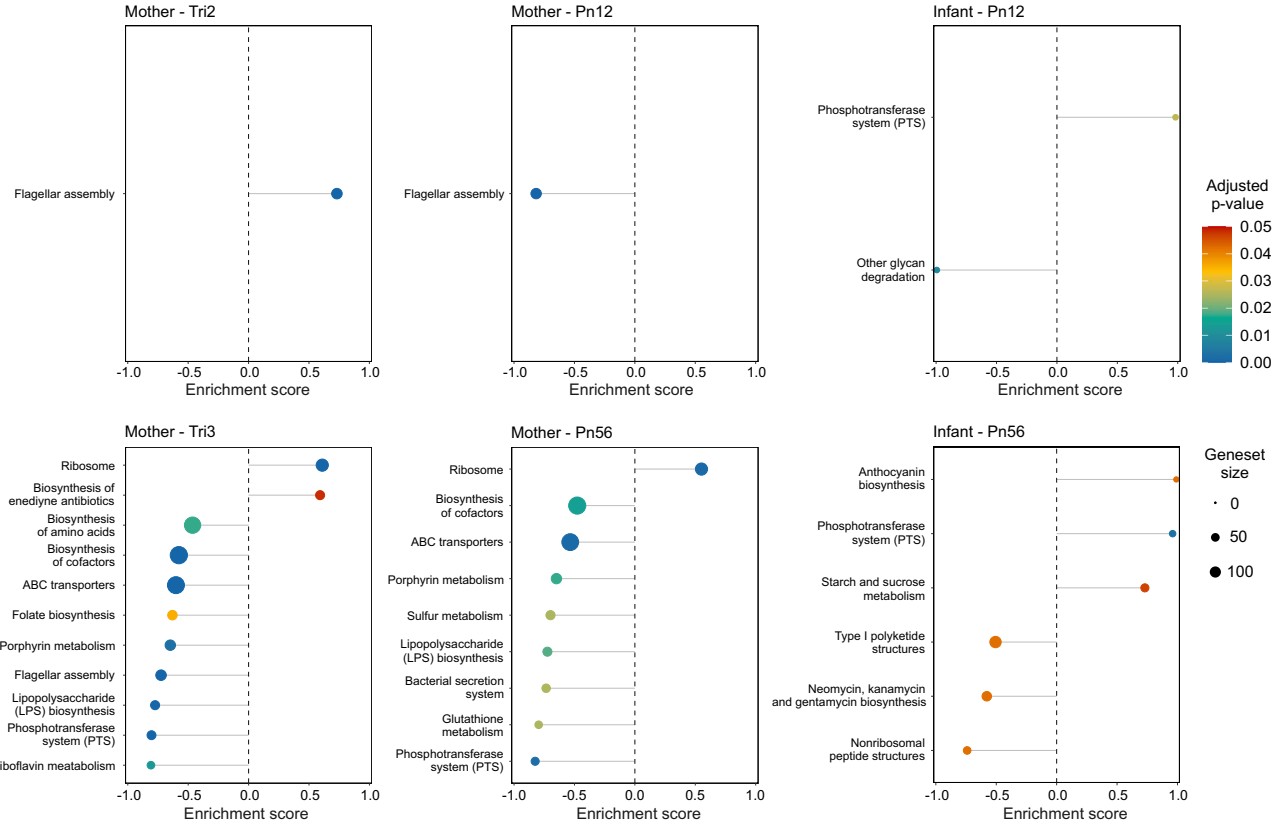

**Fig. 6 | Significantly enriched or depleted KEGG pathways associated with BEP supplementation in mothers and infants across time points.** The color of each point corresponds to the adjusted *p*-value (Benjamini-Hochberg correction), with warmer colors indicating more significant changes. The size of each point reflects the size of gene set associated with the pathway. A full table of the gene set enrichment analyses can be found in Supplementary Data 2. BEP balanced energy-protein, Pn12 postnatal 1–2 months, Pn56 postnatal 5–6 months, Tri2 trimester 2, Tri3 trimester 3.

LAZ and WAZ at 3 and 6 months, though with negligible effect size (Supplementary Data 3).

We also conducted a causal mediation analysis on gene-level data by selecting the top 500 genes. While specific genes exhibited significant mediating effects on birth outcomes and infant growth, subsequent gene set enrichment analysis did not reveal distinct, significant microbial pathways. The complete results from the causal mediation analyses are available in Supplementary Data 3.

## Discussion

In this study, we explored the impact of maternal BEP supplementation during pregnancy and lactation on the gut microbiome of mothers and their exclusively breastfed infants in rural Burkina Faso. Our findings reveal that BEP supplementation induces significant shifts in the maternal gut microbiome, with notable changes in diversity, community structure, functional pathways, and the abundance of key microbial taxa. These changes also extend to the infant gut microbiome, albeit more modestly. Causal mediation analysis uncovered key microbial taxa mediating the effects of BEP supplementation on birth outcomes and infant growth, underscoring the complex interactions between maternal supplementation, gut microbiome dynamics, and early-life development in resource-limited settings.

The effect of BEP supplementation on Shannon diversity was significant during the prenatal period but not postnatally. This may reflect the distinct physiological and microbiome dynamics between these phases. During pregnancy, the maternal gut microbiome undergoes substantial remodeling to meet the metabolic demands of fetal growth[30], and dietary supplementation may interact with these changes, amplifying microbial diversity. Postnatally, the microbiome transitions toward a more stable state over time as the body adjusts to lactation and recovery[31], potentially reducing the detectable impact of supplementation. The significant effect of BEP supplementation on gene richness at 1–2 months postpartum may reflect the dynamic shifts during early lactation, as maternal metabolism adapts to meet increased energy demands. These changes, interacting with BEP supplementation, could enhance microbial functional potential, explaining the observed increase in gene richness. As lactation stabilizes, the effect of supplementation diminishes, reducing differences at later stages.

During the second trimester, BEP supplementation was associated with a reduction in microbial modularity, a decrease in the abundance of the *Oscillospiraceae* species CAG-103 sp000432375, and the enrichment of pathways related to motility and sugar biosynthesis, such as flagellar assembly and O-antigen nucleotide sugar biosynthesis. These changes likely reflect an adaptive response to the increased nutritional intake provided by BEP supplementation. The reduction of CAG-103 sp000432375, a species recently recognized as responsive to carbohydrates[32], may indicate a shift in microbial metabolism away from carbohydrate dependence. At the same time, the depletion of harmful pathways, such as those linked to pathogenic *Escherichia coli* infection, suggests a shift toward a healthier, less-inflammatory microbial environment[33].

As pregnancy progressed into the third trimester, further microbiome adaptations to BEP supplementation were observed, including an increase in network diameter and the depletion of key biosynthetic pathways, such as ABC transporters, LPS biosynthesis, and PTS. These changes suggest that BEP supplementation enhances microbial energy allocation, likely in response to the growing nutritional demands of fetal growth. The reduction in LPS

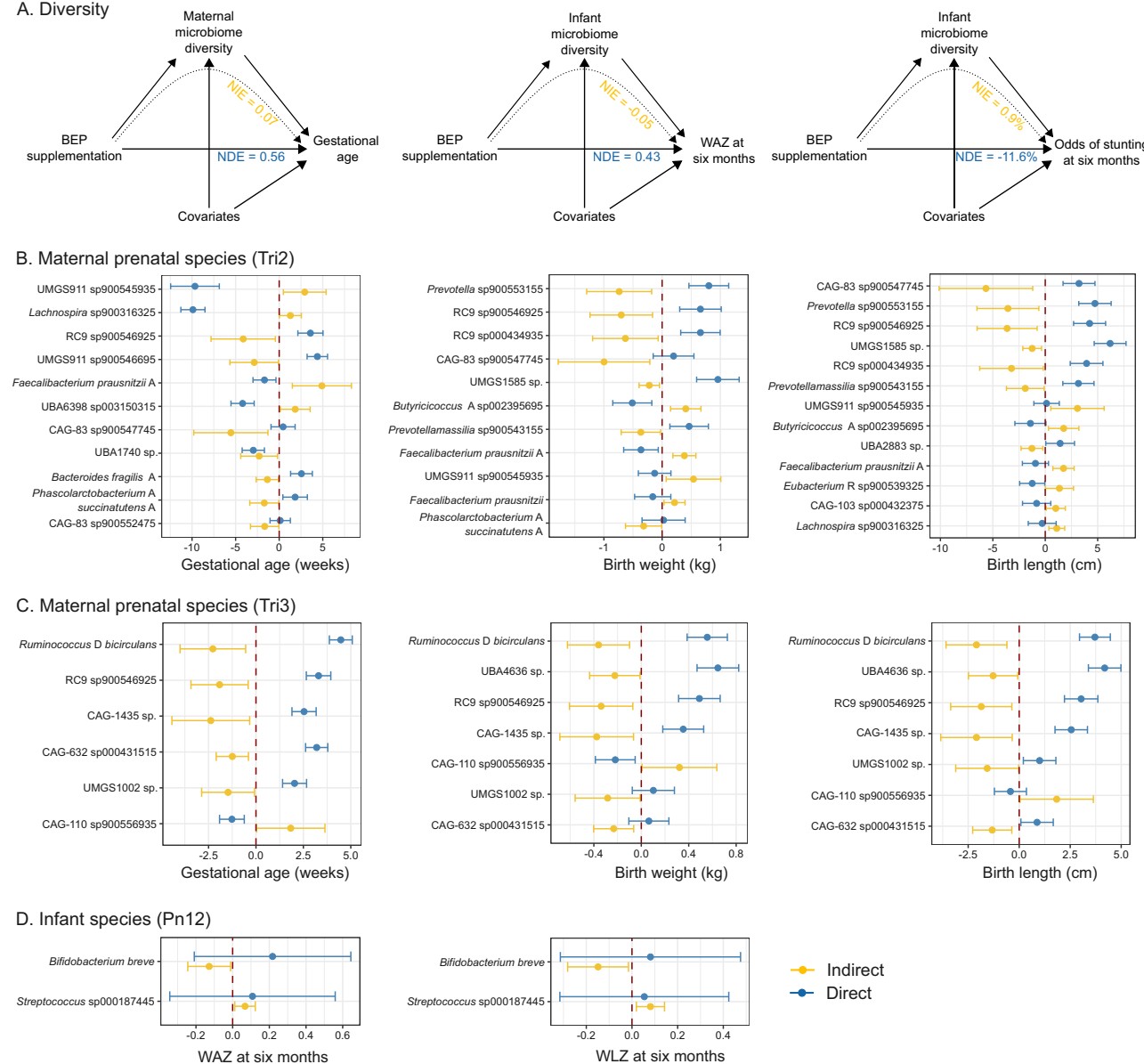

**Fig. 7 | Mediating effect of the microbiome on birth outcomes and infant anthropometry. A** Estimates for the NIE and the NDE of the microbiome diversity on gestational age at birth and on WAZ and stunting prevalence at six months. Results are superimposed on the causal directed acyclic graphs used in these analyses. The model with maternal Shannon diversity as mediator included maternal strata data as covariates (level of education, marital status, language, religion, ethnicity, number of jobs, socio-economic status, dietary diversity score, and health center), as well as maternal characteristics taken at inclusion into the trial (age, height, BMI and hemoglobin level). The model with infant microbiome diversity as a mediator also included the age at which the infant stopped exclusively breastfeeding as a covariate. Infant microbiome diversities were measured at 1–2 months of age. Estimates for the interventional direct (blue) and indirect

(yellow) effects of each individual species' abundance in the maternal microbiome in (**B**) the second trimester, (**C**) the third trimester, and (**D**) in the infant microbiome at 1−2 months of age. Data are presented as the estimated effect size from the mediation analysis, and the lower and upper bounds of the 95% confidence interval on the effect estimate. Maternal Tri2 samples: Gestational age ($n = 42$); Birth weight ($n = 41$); Birth length ($n = 41$). Maternal Tri3 samples: Gestational age ($n = 147$); Birth weight ($n = 143$); Birth length ($n = 143$). Infant Pn12 samples: WAZ at six months ($n = 121$); WLZ at six months ($n = 121$). The complete results of these mediation analyses are available in Supplementary Data 3. NIE Natural Indirect Effect, NDE Natural Direct Effect, BEP balanced energy-protein, Tri2 trimester 2, Tri3 trimester 3, Pn12 postnatal 1–2 months, WAZ weight-for-age Z-score, WLZ weight-for-length Z-score.

is particularly relevant, as its pro-inflammatory properties have been linked to preterm labor[34]. Furthermore, the reduced reliance on carbohydrate metabolism during pregnancy is supported by the concerted depletion of genes for the PTS and ABC transporters observed in the metagenomes of BEP-supplemented participants in the third trimester[35,36]. Previous studies have shown that a microbiome's energy extraction efficiency from diet correlates with an enrichment of specific metabolic pathways, particularly those for carbohydrate transport and utilization[30,37]. These findings suggest

that BEP supplementation optimizes maternal energy storage to support fetal development.

Postnatally, BEP supplementation continued to influence the maternal gut microbiome, with distinct patterns compared to the prenatal period. The intervention group exhibited higher modularity, suggesting more compartmentalized microbial communities. Increased modularity may reflect a shift toward a more specialized network structure, where microbial taxa form distinct and cohesive clusters with limited interaction between groups. This

compartmentalization can enhance the resilience of the microbiome to environmental changes, potentially safeguarding key functional groups from external disturbance[38]. This was accompanied by an increased abundance of *B. fragilis* and a persistent depletion of key biosynthetic pathways related to carbohydrate metabolism and pro-inflammatory responses. *B. fragilis* is known for releasing bacterial polysaccharides that direct the maturation of the immune system[39,40]. The observed depletion of inflammatory pathways, such as LPS bio-synthesis, supports the idea that BEP supplementation plays a pro-tective role by promoting a more stable and less inflammatory postnatal environment[41]. These findings suggest that BEP supple-mentation not only helps compartmentalize commensal microbiota but also supports maintaining a homeostatic microbiota composition, which is crucial for maintaining host immunity[42].

In contrast to the pronounced changes in mothers, the infant gut microbiome showed more modest responses to maternal BEP sup-plementation. However, infants born to BEP-supplemented mothers exhibited more rapid changes in their gut microbiome composition over time, suggesting greater microbial adaptation during early life. Functionally, maternal BEP supplementation was associated with enriched carbohydrate metabolism pathways in the infant gut, such as PTS and starch metabolism, potentially influenced by changes in breast milk composition. Notably, the involved genes are cellobiose-related, which have been reported to transport lactose[43,44]. In neonates and young infants, up to 20% of dietary lactose may reach the colon, where it is fermented by the colonic microbiota[45]. While the con-centration of lactose in human milk remains relatively stable, mothers producing higher milk volumes tend to have elevated lactose concentrations[46]. Therefore, BEP-supplemented mothers may produce higher milk volumes, providing more lactose for infant growth and potentially influencing the carbohydrate metabolic pathways in the infant gut microbiome.

Our mediation analysis revealed that BEP supplementation had significant direct effects on birth outcomes, including gestational age, birth weight, and length. These effects were mediated by specific microbial taxa rather than overall community-level changes. Notably, certain *Prevotella* species and members of the order Bacteroidales, such as RC9 sp900546925, showed decreased abundances in mothers after the intervention, which were estimated to negatively correlate with gestational age, birth weight, and length. In contrast, species from the order Oscillospirales, such as CAG-110, *Faecalibacterium*, and *Butyricicoccus*, were estimated to have positive changes in abundance with respect to these birth outcomes. In infants, taxa like *B. breve* and *Streptococcus* sp000187445 exhibited significant mediating effects on WAZ and WLZ at six months. However, these effects were modest, indicating that while the microbiome contributes to infant growth, other factors like maternal nutrition and overall health may play a more dominant role. While we recognize that the counterfactual nat-ure of indirect effects makes them difficult to interpret, these analyses highlight species that play a key regulatory role in their ecosystem. Diverging indirect and direct effects, such as seen for *Ruminococcus* D *biciruclans* in the third trimester, indicate a regulating role, counter-acting effects from the intervention and other microbiota. Conversely, indirect effects that outweigh or compound the direct effects highlight changes in birth outcomes that are mechanistically caused by specific microbiota.

Our study has several strengths and limitations. It presents a rare combination of deep sequencing depth and a relatively large sample size from a population that is underrepresented in metagenomic literature[47]. Samples were collected at multiple time points from mother-infant dyads, covering prepartum and postpartum periods, allowing us to capture the natural changes in microbial composition[48]. Furthermore, compliance with BEP and IFA supplementation was verified by a community-based network of home visitors, resulting in high levels of observed adherence. However, several limitations

warrant caution. First, this study was launched after the MISAME-III trial recruitment had already completed, with most participants already in their third trimester. Therefore, the sample size for the second trimester was smaller than subsequent time points, and women had been receiving BEP supplementation for several weeks, which limited the ability to access the prenatal gut microbiome prior to supplementation. Second, the absence of inflammation bio-markers, such as C-reactive protein and $\alpha_1$-acid glycoprotein, limited our ability to directly link microbial shifts with inflammatory states. Finally, some differential pathways, such as those for *Escherichia coli* infection, the ribosome complex, biosynthesis of cofactors, and porphyrin metabolism, are unlikely to be of microbial origin. This may be a limitation of the KEGG database used for gene annotation, but its consistent association with the BEP-supplemented metagenome may indicate the presence of unknown microbial processes involving these genes.

In conclusion, our findings shed light on gut microbiome dynamics in an underrepresented population. Maternal BEP supple-mentation during pregnancy and lactation significantly reshapes the maternal gut microbiome, while exerting more modest yet detectable effects on the infant microbiome. The intervention alters microbiome diversity, community structure, and function in the maternal gut and enhances the carbohydrate metabolic capacity in the infant's gut. These shifts may benefit maternal metabolic and immune health while promoting fetal and neonatal growth. Future studies focusing on human milk and incorporating multi-omics approaches, such as pro-teomics and metabolomics, will be essential to fully elucidate the broader physiological effects of maternal supplementation, particu-larly in resource-limited settings.

## Methods

Written informed consent was obtained for all subjects, including parents or legal guardians when applies.

### Study setting and design

The biospecimen sub-study (BioSpé)[25] was nested within the framework of the MISAME-III trial, which evaluated the physiological effects of BEP supplementation on pregnant and lactating women and their infants in Burkina Faso. The MISAME-III study protocol[49] was published previously. In brief, it was an individually randomized 2 × 2 factorial controlled trial evaluating the efficacy of BEP supplementation, fortified with multiple micronutrients, during pregnancy and lactation on birth outcomes and infant growth. The study was conducted across six rural health centers in the district of Houndé, Hauts-Bassins region, Burkina Faso.

Eligibility criteria for the trial included confirmed pregnancy by a urinary test and ultrasound examination and signed informed consent. Women with a gestational age ≥21 completed weeks, planning to leave the study area, or allergic to peanuts were excluded. In total, 1897 women aged between 15 and 40 years were enrolled from October 30, 2019 to December 12, 2020. The last child was born on August 7, 2021. These women were randomly assigned to one of two prenatal inter-vention arms, receiving either BEP supplementation with IFA tablets (i.e., intervention) or IFA alone (i.e., control). They were further ran-domized into one of two postnatal arms to receive either BEP sup-plementation during the first six months postpartum in combination with IFA for the first six weeks (i.e., intervention), or IFA alone for six weeks postpartum (i.e., control). Thus, the MISAME-III study included four groups: (1) prenatal BEP and IFA supplementation; (2) postnatal BEP and IFA supplementation; (3) both pre- and postnatal BEP and IFA supplementation; and (4) both pre- and postnatal IFA-only supplementation.

Maternal and infant anthropometric measurements, household and maternal baseline characteristics, infant feeding practice, and clinical assessments were collected by study physicians and midwives using SurveySolution (version 21.5) on tablets.

The BioSpé study was initiated following the completion of recruitment for the MISAME-III trial, when most participants were in their third trimester. To maximize recruitment of participants in their second trimester, women were prioritized based on gestational age in descending order, ensuring even distribution across all four study groups. Ultimately, 309 women and their infants were enrolled into the BioSpé study, and various biospecimen samples were collected at multiple time points. Here, we report the metagenomic analysis of fecal samples from women and their infants. Fecal samples were collected from mother-infant dyads across all four trial arms, but only two groups of samples ($n = 152$) were sequenced due to limited funding: 71 pairs from the intervention group (receiving both pre- and postnatal BEP and IFA supplementation) and 81 pairs from the control group (receiving both pre- and postnatal IFA tablets only). Maternal fecal samples were collected at four time points: the second trimester (19–24 weeks), when women had been supplemented with BEP for approximately 12 weeks; the third trimester (30–34 weeks); 1–2 months postpartum (28–56 days); and 5–6 months postpartum (147–175 days). Infant fecal samples were collected at 1–2 months (12–56 days) and 5–6 months (140–168 days) after birth, corresponding to maternal postpartum time points.

## Study supplements

The BEP supplement used in this trial was a ready-to-consume, lipid-based nutrient supplement in the form of an energy-dense peanut paste fortified with multiple micronutrients. The supplement, produced by Nutriset, does not require a cold chain during transportation, is highly stable, and has a long shelf life. The complete nutritional composition of a daily dose of the fortified BEP is provided in Table S2. The daily dose (72 g) provided 393 kcal, with 36% energy from lipids, 20% from protein, and 32% from carbohydrates. The protein came from a combination of soy (61%), milk (25%), and peanut (15%). The daily dose covered at least the estimated average requirements of pregnant women for 11 micronutrients, but not calcium, phosphorus, and magnesium[50].

In addition to BEP supplements, all women in the study received IFA tablets, which are part of the standard prenatal care in Burkina Faso. The daily dose of an IFA tablet (produced by Sidhaant Life Science, Delhi, India) contained 65 mg of iron [form: $FeH_2O_5S$] and 400 μg of folic acid [form: $C_{19}H_{19}N_7O_6$]. Following Burkinabè guidelines, all enrolled women received malaria prophylaxis, consisting of three oral doses of sulfadoxine-pyrimethamine.

## Fecal samples collection procedure

Maternal fecal samples (8 g) were collected using stool collection containers, aliquoted into sterile cryotubes (Biosigma, Cona, VE, Italy), and immediately flash-frozen before being transferred to a −80 °C freezer. Infant feces (8 g) were collected using a 38 × 50 cm sterile protection sheet (Kimberley-Clark, Irving, TX, USA) wrapped around the newborn, which functioned like a diaper. After collection, infant feces were transferred to an OMNIgene•GUT OM-200 collection kit and stored at −80 °C.

Fecal consistency was assessed using the visual Bristol scale. In the case of liquid feces, samples were thoroughly homogenized using a plastic spoon to ensure uniformity between solid and liquid components. Throughout transportation, fecal samples were kept at −80 °C using dry ice, and the temperature was monitored with a logger to ensure stability. Samples were stored at this temperature until they were transported to the laboratory for further analysis.

## Metagenomic library preparation and sequencing

DNA extraction, library preparation, and metagenomic sequencing were performed at the UC San Diego IGM Genomics Center. DNA extraction was performed using the Thermo MagMAX Microbiome Ultra kit, and DNA quantification was performed using the Pico-Green dye. Roche KAPA HyperPlus kits and 96-UDI plates were used for library preparation. In this study, 25 ng of input DNA and nine PCR cycles were used during library preparation. A shallow MiSeq Nano sequencing run was used to normalize read sequencing depth. Subsequently, 2 × 150 bp reads were generated on the Illumina NovaSeq 6000 at a target depth of 25 gigabase pairs per sample for maternal samples and 10 gigabase pairs per sample for infant samples.

## Metagenomic read annotation

The metagenomic sequencing reads were trimmed and deduplicated using FastP. All reads mapping to the hg38 human genome with Bowtie2 under default settings were removed. Samples were profiled by mapping preprocessed reads from each sample to a previously established database of microbiome genomes (database "DeltaI") using Bowtie2 with default settings. The DeltaI microbial genome database was specifically created to profile non-industrialized infant microbiomes[17]. The resulting mapping files were analyzed using inStrain profile under default settings[51]. The relative abundance of each genome was calculated using the formula: number of reads mapping to the ratio of genome and number of processed reads in the sample. Only genomes with breadth > 0.5 were considered in analyses.

Gene-level functional profiling was performed using the inStrain "parse_annotations" pipeline with default settings (https://instrain.readthedocs.io/en/master/user_manual.html#parse-annotations). Genes were called for each genome individually using Prodigal in "single" mode, and all gene files were then concatenated together. We performed functional annotation against (1) KEGG Orthologies[52] using kofam_scan (version 1.3); (2) a carbohydrate-active enzymes database[53] (CAZymes, http://www.cazy.org/) using dbCAN2 (version 11); (3) a Comprehensive Antibiotic Resistance Database (CARD) Anti-Microbial Resistance database[54] version 3.2.5 using Diamond with the command "diamond blastp -f 6 -e 0.0001 -k 1"; and (4) human milk oligosaccharide (HMO) proteins using the instructions provided here: https://instrain.readthedocs.io/en/master/user_manual.html#human-milk-oligosaccharide-hmo-utilization-genes.

## Statistical analysis

The statistical analysis was performed in R (v4.3.1).

**Alpha and beta diversities.** Alpha-diversity metrics, including the Observed Species index, Simpson's Diversity index, and Shannon Diversity index, were calculated using *phyloseq* (v1.44.0)[55]. Treatment effects of the randomized intervention were estimated using the doubly robust TMLE with the *tmle* package (v1.5.0)[56]. To account for the heterogeneity of effects between populations, these analyses were performed separately for maternal and infant data and further separated into pre- and postnatal for the maternal samples. Maternal stratum data (level of education, marital status, language, religion, ethnicity, number of jobs, socio-economic status, dietary diversity score, and health center), as well as anthropometric data taken at inclusion into the trial (age, height, BMI) and hemoglobin level, were included as covariates for the model. Both covariate-outcome and propensity score models were fitted with ensemble SuperLearner (v2.0-28). Diversity indices were translated to effective numbers of species using the formulas proposed by Jost[57].

Beta-diversity analysis was performed using the *vegan* package (v2.6-4). Given the phylogenetic structure of taxonomic data, we used weighted UniFrac distances calculated from log-transformed strain counts for ordination analysis. For gene data, which lack a clear phylogenetic structure, we applied Bray-Curtis distances after normalizing the gene counts using total sum scaling, followed by a log

transformation to reduce the effects of high-abundance genes and stabilize variance. Group comparisons were assessed using permutational multivariate analysis of variance (PERMANOVA) with default parameters.

**Differential abundance analysis.** To address the compositionality and zero-inflation issues of microbial count data, the differential abundance analyses were performed using ANCOM-BC2 (v2.2.1) at the species level. Analyses were conducted at both combined time points and individual time points, excluding the second trimester due to limited sample availability. In the combined time point analysis, the individual subject identifier was included as a random effect to account for repeated sampling of participants. Benjamini-Hochberg multiple testing correction was used, and the alpha level was set at 0.05. ANCOM-BC2 handles zero-inflation by adding pseudo-counts to the data, which can considerably influence the false-positive and false-negative rates[58]. To mitigate this concern, the ANCOM-BC2 package conducts a sensitivity analysis to evaluate the effect of varying pseudo-counts (ranging from 0.01 to 0.5 in increments of 0.01) on zeros for each taxon. If a taxon is found to be sensitive to pseudo-counts, then it is declared as a non-significant taxon[58]. Therefore, only robustly significant taxa were included in this analysis.

To assess the bacterial changes across time points between the intervention and control groups, a mixed-effects linear regression model was utilized. These analyses focused on the top ten taxa that were present at all time points with a minimum abundance of 1%. Data from the second trimester were excluded from these analyses due to the limited number of samples collected at that time point.

**Co-occurrence network analysis.** To investigate microbial interactions and co-occurrence patterns within the maternal and infant gut microbiomes, we conducted a co-occurrence network analysis using strain-level data. Microbial associations were inferred using the *SpiecEasi* package (v1.1.3) for ecological association inference[59]. For robustness, taxa with an abundance lower than 1% were filtered, retaining only those present in at least 5% of the samples. Networks were constructed for both the intervention and control groups at each time point, with nodes representing microbial taxa and edges reflecting significant co-occurrence relationships. Microbial community structures within the networks were identified using the Louvain community detection[60] via the *igraph* package (v2.0.3)[61].

**Functional data analysis.** Gene-level functional profiling was conducted using hierarchical differential abundance analysis on the trimmed mean of M-values (TMM) library sizes for the gene counts using the *treeclimbR* (v0.1.5) and *edgeR* (v3.42.4) packages[62]. Genes from KEGG, HMO, CAZyme, and CARD were categorized into a hierarchical structure based on their respective enzymatic class number, functional group, HMO cluster or antibiotic resistance mechanism. Benjamini-Hochberg multiple testing correction was used, and the alpha level was set at 0.05. To identify pathways that were differentially abundant in a coordinated direction, gene set enrichment analysis was performed using KEGG pathway mappings[63]. Additionally, we included a *Bifidobacterium longum* NagR regulon gene set from Arzamasov et al. [64] for HMO utilization genes. For each set, an enrichment score with a corresponding adjusted *p*-value was calculated using the *clusterProfiler* package (v4.2.2)[65]. The alpha level was set at 0.05, and the minimum gene set size was set to two.

**Mediation analysis.** Mediation analysis was performed to examine the mediating effect of microbial diversity and individual taxa on the effect of BEP supplementation on a number of birth outcomes and anthropometry at three and six months. The natural direct and indirect effects of microbial diversity were estimated using TMLE

models implemented in the *medoutcon* package (v0.2.0)[66]. For binary outcomes (low birth weight, SGA, and stunting prevalences at six months), binomial g- and b-learners were used with Gaussian h- and r-learners from *sl3* (v1.4.5)[67]. For continuous outcomes (infant weight, length, and gestational age at birth, WAZ, WLZ, and LAZ at three and six months), binomial g-learners were used with Gaussian h-, b-, and r-learners. The interventional direct and indirect effects of individual taxa as mediators were estimated using the *HDmediation* package (v0.1.0.9000)[68] using the abundances of all other taxa as potential intermediate confounders. For continuous outcomes, effect size differences were estimated, while for binary outcomes, log-odds ratios were estimated.

### Reporting summary
Further information on research design is available in the Nature Portfolio Reporting Summary linked to this article.

## Data availability
The metagenomic sequencing data generated in this study have been deposited in the NCBI database under BioProject accession code PRJNA1113165. The de-identified personal data are available under restricted access for the purpose of verifying, interpreting, or extending the research published in this article. Access can be obtained by contacting carl.lachat@ugent.be and signing a data-sharing agreement. The processed taxonomy and gene data are available at https://github.com/stefftaelman/misame-metagenomics-BEP. A response to requests for data access can be expected within two months, and the data will be available for one year after signing the data-sharing agreement.

## Code availability
The R code for data analysis and source data are available at Zenodo https://doi.org/10.5281/zenodo.14918417 and GitHub https://github.com/stefftaelman/misame-metagenomics-BEP.

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

## Acknowledgements

The work is supported by the Bill & Melinda Gates Foundation (OPP1175213). Lishi Deng is supported by the China Scholarship Council (Grant No. 202207650056). Steff Taelman is supported by the Flemish Agency for Innovation and Entrepreneurship (VLAIO HBC.2020.2292). This publication includes data generated at the UC San Diego IGM Genomics Center utilizing an Illumina NovaSeq 6000 that was purchased with funding from a National Institutes of Health SIG grant (#S10 OD026929). The funders had no role in the design and conduct of the study; in the collection, management, analysis, and interpretation of the data; or in the preparation, review, or approval of the manuscript. The authors thank all participants from Burkina Faso and the data collection team. We thank Nutriset (France) for donating the BEP supplements.

## Author contributions

L.D. and S.T. performed the statistical analyses, visualized data, interpreted data, and prepared the manuscript with contributions from all authors. M.R.O., E.D.S., and J.L.S. were responsible for the sample analysis, interpreted preliminary data and critically reviewed the manuscript. T.D.C., L.C.T., C.L., L.H. and P.K. conceived of the study, acquired funding, and critically reviewed the manuscript. T.D.C., L.C.T., L.O.O., Y.B.M., M.O., R.G., C.L., L.H. and P.K. set up the study and supervised the project. T.D.C., L.O.O., Y.B.M., M.O. and R.G. collected samples. T.D.C., E.B., A.A., K.T., W.V.C., and M.S. supervised the data curation and contributed to data interpretation. All authors have read and agreed to the published version of the manuscript.

## Competing interests
The authors declare no competing interests.

## Inclusion & Ethics statement
The study was approved by the Commissie voor Medische Ethiek (CME) of Ghent University Hospital (protocol code: B670201734334 and date of 10/08/2020) and the Comité d'Éthique Institutionnel de la Recherche En Sciences de la Santé (CEIRES) of the Institut de Recherche en Sciences de la Santé (IRSS) (protocol code: 50-2020/CEIRES and date of 22/10/2020). An independent Data and Safety Monitoring Board (DSMB), which included an endocrinologist, two pediatricians, a gynecologist, and an ethicist from Belgium and Burkina Faso, was established prior to the trial. The DSMB managed remote safety reviews for adverse and serious events at nine and 20 months after the start of enrollment. The MISAME-III trial was registered on ClinicalTrials.gov (identifier: NCT03533712). The research was conducted in collaboration with local partners, including Agence de Formation de Recherche et d'Expertise en Santé pour l'Afrique (AFRICSanté) and IRSS. Researchers from AFRICSanté and IRSS who contributed to the research and fulfilled the authorship criteria were included as co-authors, while other team members, such as physicians and midwives who contributed to data collection, were acknowledged in the Acknowledgements section. Roles and responsibilities were agreed upon among collaborators ahead of the research initiation. A clear delineation of tasks and contributions was outlined in collaborative meetings and documented in written agreements to ensure effective coordination and execution of the study objectives. Additionally, capacity-building plans for local researchers were discussed and integrated into the research framework, including training sessions, knowledge-sharing initiatives, and authorship opportunities. The research does not lead to stigmatization, incrimination, discrimination, or personal risk for participants. Thorough risk assessments were conducted prior to the commencement of the research to identify and evaluate potential hazards and risks associated with various aspects of the study. These assessments considered factors such as environmental conditions, exposure to hazardous materials, and travel risks. Benefit-sharing measures have been discussed in the event that biological materials, cultural artifacts, or associated traditional knowledge are transferred outside the country. Local and regional research relevant to our study has been taken into account in our citations.

## Additional information

Lishi Deng[1,14], Steff Taelman [2,3,4,14], Matthew R. Olm [5,6], Laeticia Celine Toe [1,7], Eva Balini[4], Lionel Olivier Ouédraogo [1,8], Yuri Bastos-Moreira[1,9], Alemayehu Argaw[1], Kokeb Tesfamariam[1], Erica D. Sonnenburg [5], Giles T. Hanley-Cook[1], Moctar Ouédraogo [10], Rasmané Ganaba[10], Wim Van Criekinge [2], Lieven Huybregts[1,11], Michiel Stock [3], Patrick Kolsteren[1], Justin L. Sonnenburg[5,12,13], Carl Lachat [1] ✉ & Trenton Dailey-Chwalibóg [1,10] ✉

[1]Department of Food Technology, Safety and Health, Faculty of Bioscience Engineering, Ghent University, 9000 Ghent, Belgium. [2]BIOBIX, Department of Data Analysis and Mathematical Modelling, Ghent University, 9000 Ghent, Belgium. [3]KERMIT, Department of Data Analysis and Mathematical Modelling, Ghent University, 9000 Ghent, Belgium. [4]BioLizard nv, 9000 Ghent, Belgium. [5]Department of Microbiology and Immunology, Stanford University School of Medicine, Stanford, CA, USA. [6]Department of Integrative Physiology, University of Colorado Boulder, Boulder, CO, USA. [7]Institut de Recherche en Sciences de la Santé (IRSS), Unité Nutrition et Maladies Métaboliques, Bobo-Dioulasso, Burkina Faso. [8]Centre Muraz, Bobo-Dioulasso, Burkina Faso. [9]Center of Excellence in Mycotoxicology and Public Health, MYTOXSOUTH® Coordination Unit, Faculty of Pharmaceutical Sciences, Ghent University, 9000 Ghent, Belgium. [10]Agence de Formation de Recherche et d'Expertise en Santé pour l'Afrique (AFRICSanté), Bobo-Dioulasso, Burkina Faso. [11]Nutrition, Diets, and Health Unit, International Food Policy Research Institute (IFPRI), Washington, DC, USA. [12]Chan Zuckerberg Biohub, San Francisco, CA, USA. [13]Center for Human Microbiome Studies, Stanford University School of Medicine, Stanford, CA, USA. [14]These authors contributed equally: Lishi Deng, Steff Taelman. ✉e-mail: carl.lachat@ugent.be; Trenton@Dailey-Chwalibog.com

