## [Transparent Peer Review file · Nature Communications]

Maternal balanced energy-protein supplementation reshapes the maternal gut microbiome and enhances carbohydrate metabolism in infants: a randomized controlled trial

Corresponding Author: Dr Trenton Dailey-Chwalibóg

Version 0:

Reviewer comments:

Reviewer #1

(Remarks to the Author)

In this study, Deng and colleagues analyse the gut microbiome of women and their infants enrolled in a randomized controlled trial of maternal balanced energy-protein supplementation in Burkina-Faso. They report subtle differences in the gut microbiome of supplemented mothers and their infants and report that beneficial effects of BEP supplementation on infant growth are mediated by the gut microbiome. This paper reports an interesting dataset from an underrepresented geographical region with respect to human microbiomes. The results are therefore novel, however greater clarification is required on the analyses and results as outlined below. Major points include clarification of grouped timepoints for analyses and comparisons of timepoints versus groups in addition to expansion of the mediation analyses beyond taxonomic diversity.

- Abstract: Indicate when BEP was provided i.e. during and after pregnancy?
- Line 15-16: This sentence doesn't necessarily support the citation used. This citation looks at 1 month diversity and rapid growth in the context of an obesogenic environment. Therefore I would disagree with what the authors say that a more mature gut microbiome in the first month may improve infant growth trajectories. Rapid growth isn't always a good thing. I would change this to "influence" infant growth or provide context to say that this study is in a high income setting prone to overweight/obesity
- In my opinion, the taxonomic tree in Fig 2B doesn't provide any useful information that adds to the results more than what is already presented in Fig 2A. I suggest it is removed/replaced
- How were participants selected for biospecimen inclusion? Random?
- Figure 2C mother groups all labelled as "postnatal". Should 2 of these be "prenatal"? Also, it is not clear if these include multiple timepoints grouped together? It would be more suitable to separate out each time point as per other figures
- Line 91 – "on average higher" – was this significant? Please provide p values etc.
- Line 92-93 – please clarify which time points are being compared when stating that there was a significant increase in Shannon diversity. It is not clear from the figure or text between which groups/timepoints there is a significant effect on diversity
- Line 95-97: why are infant postnatal visit collapsed together? The Shannon diversity at 2 weeks would be expected to be very different to that at 8 weeks.
- Figure 2D/E caption colours are wrong
- Clarification on the addition and use of pseudo-counts?
- Figure 3: Can you provide a figure with boxplots and individual datapoints for *B. fragilis* and the *Oscillibacter* CAG relative abundance? This would be helpful to visualize the difference in abundances between groups at the relevant timepoints
- Overall, the separation of "prenatal" and "postnatal" results makes things a little confusing, as it is clearly indicated in the methods that only the combined prenatal+postnatal intervention groups were included in this study
- In the mediation analysis, more detail is needed on the direction of effects in lines 115-118. It seems from figure 2 that BEP, if anything, decreases infant diversity, albeit not significantly, suggesting the mediation effect is via reducing Shannon diversity. I think it would be helpful to expand figure S5 but to include the NIE and NDE values on each arrow/interaction to show the strength and direction of each effect. This figure could be put in the main text as it is central to the story.
- Please also clarify the "timepoint" in table S2 to confirm if the mediation is with combined timepoints.

- Major point: Due to the major limitation of no true baseline sample, it is important to indicate in all results what the comparison group is when indicating a significant change in diversity/abundance etc. For example, in lines 131-147 please indicate if the “enrichment” of genes in BEP-supplemented mothers is in comparison to earlier timepoints in the same mothers or in comparison to non-supplemented mothers at the same time point.
- Line 157-158: The “striking disparity” between mothers and their infants wouldn’t usually be considered a “novel finding” as this has been reported in every dataset of maternal-infant microbiomes around the world.
- Major point: Mediation analysis is currently only for taxonomic diversity and not for other things such as individual pathways/taxa or PC1/PC2 score. Can you expand the mediation analysis to include some of these things?
- Supplementary tables need a key to explain what each column header means.
- It may be useful to check metagenomic diversity (gene richness), its changes with supplementation and it’s mediating effect. Similarly, it would be beneficial to look at functional B-diversity as most of the effects were in functional data.

Reviewer #2

(Remarks to the Author)

Dear Authors,

Thank you for providing a well written manuscript in a population which is understudies. I will provide general feedback on the four main outcomes outlined in the discussion followed by more specific feedback.

The main findings are:

1- Striking disparity in the gut microbiome composition between mothers and their infants.

This is not a novel finding, multiple papers have found this and the infant microbiome is distantly different from the adult gut microbiome. This should be acknowledged and the finding should not be featured as novel.

2- BEP supplementation affected specific facets of the maternal gut microbiome, increase in prenatal microbial diversity, with a decrease in the abundance of an Oscillospiraceae species during pregnancy and an increase of *Bacteroides fragilis* at 5-6 months postpartum.

BEP supplementation is a significant addition to the diet (72 g per day) and it would therefore be expected to result in changes to the maternal gut microbiome. Because of the population group the effect of the supplement is likely to have a greater effect in this population compared to for example a Western population. The change is interesting and maybe novel for this population group and with this intervention, but not unexpected. A diet supplementation should be seen in relation to the habitual diet of the individual. However, no diet information is given in the manuscript it's therefore not possible for the reader to evaluate the impact of the supplement on diet intake. It would be interesting to know how much the supplement increase the protein intake as that will affect the carbohydrate/protein ratio which is very important for changing the gut microbiome.

3- Mediation analysis revealed that BEP supplementation drives infant growth through microbiome diversity.

The evidence provided doesn't convince me that it's the microbiome diversity that is driving the growth of the infant as it's likely that a mother having additional food (BEP) produces more milk and therefore the infant thrives better. Therefore, the change in microbiome diversity is a secondary effect and not the driver of the effect. It should also be kept in mind that no difference in beta-diversity was found between supplementation and control group.

4- Multiple microbial pathways associated with BEP supplementation, of which, the PTS genes for carbohydrate uptake were enriched in infant samples whose mothers were BEP-supplemented

Could this be related to the adaptation of the microbiome to changes in the infant gut related to better maternal nutrition affecting the infant through breastfeeding?

I did notice that the initial maternal sampling time had a reduced number of samples due to the sample collection starting after the main study. However, for the remaining timepoint it would have been interesting to know how many mother-infant dyads had provided samples at all time points. This could have allowed for a repeated measure analysis which could remove some of the variability between timepoints.

L 98 - States that infant had between 2 and 43 species. I would question if a sample with only 2 identified species is a true representation of the microbiome.

Overall, I find the research interesting but the outcome is as expected.

Reviewer #3

(Remarks to the Author)

Deng et al. present an analysis of data collected from mother-infant dyads in a randomized controlled trial conducted in rural Burkina Faso. The trial aimed to study the effects of balanced energy-protein (BEP) supplementation on the gut microbiome. This data is unique, primarily due to metagenomic sequencing and a relatively large sample size from an underrepresented population. While the data collected are impressive and certainly serve as a valuable resource, there are multiple shortcomings in the analyses conducted. In particular, the treatment of the statistical analysis is unfocused and includes sporadic information without clear direction. A more effective approach would have been concentrating on fewer key results, analyzing and presenting in greater depth. As a result, the overall impact is diluted. My specific comments are as follows:

1. Although the study had unique access to mother-infant dyads at multiple time points, none of the analyses considered jointly modeling the mother-child pair as a single analytical unit, as is common practice in maternal-fetal human genetics studies. This oversight represents a significant missed opportunity to leverage the potential of the data fully.
2. The availability of longitudinal samples at both pre- and post-intervention time points provided a unique opportunity for dynamic analysis. However, no assessment of temporal variability in taxonomic and functional composition was described. As a result, the study failed to capitalize on the time-varying aspect of gut microbiome changes, which diminishes the depth of the findings and overlooks potential insights into the developmental trajectories of the microbiome.
3. The major conclusions of the paper focus solely on microbiome diversity. While diversity is a useful metric, it is only one aspect of understanding an ecosystem's complexity. Current literature indicates that microbiome diversity alone does not provide a complete picture and must be considered alongside other factors such as stability, structure, and function. This is particularly relevant, as comparisons of diversity between two clinical groups often yield contradictory results. Therefore, a broader analytical approach, based on specific microbiome features (beyond what is conducted in the manuscript), is necessary to comprehensively support the conclusions.
4. The statistical analysis in general has been quite ordinary, especially for a journal of this caliber. There are probably too many issues to mention, but I will summarize the three key ones here. First, ANCOM-BC was used for differential abundance analysis, a method that has consistently exhibited a high empirical FDR in recent benchmarking studies (PMIDs: 35421994, 38701410). Moreover, a recent highly cited publication highlighted the inconsistency of current differential abundance methods (PMID: 35039521), indicating that reliance on only one method is risky. It should not be too difficult to run multiple methods to identify the most consistent signals in the data.
5. Second, the follow-up analysis of the differential abundance analysis did not include an enrichment analysis, which was surprising as it would have been more appropriate to investigate the microbial functional pathways based on the differential abundance analysis results.
6. Finally, the mediation analysis was surprisingly limited to microbial diversity alone, despite the availability of multiple tools for conducting per-feature mediation analysis (individual microbial features as mediators). Such an analysis might provide more comprehensive and persuasive evidence to support the established causality.
7. A minor comment to conclude my comments on the statistical methods: the treatment effect of the randomized intervention was estimated using the doubly robust TMLE. What are the components of the multiple SuperLearner models included in this exercise? How sensitive are the results concerning various combinations of base models? Given that there are so many parameters involved, I was surprised to see no comprehensive sensitivity analysis of hyperparameters is conducted which is a major drawback of this particular analysis.

In summary, despite collecting a rich dataset across time points, treatment groups, and experimental units, the analyses fall short in many ways in generating solid, convincing biological interpretations. Unfortunately, it would take quite a bit of work to complete the study. Without a major comprehensive re-analysis, it is not clear if this manuscript serves as a meaningful contribution to justify publication in Nature Communications.

Version 1:

Reviewer comments:

Reviewer #1

(Remarks to the Author)

I thank the authors for responding to most of the comments suitably. As the manuscript is considerably altered following the first version, I have further comments that need to be addressed below. In general, I am not sure that some of the additional figures/analyses are necessary outside of what was proposed in the original reviewers comments. Simple, clear figures showing the most important results should be saved for the main text. Anything else should be saved for supplementary. I

appreciate that there are many detailed suggestions, however these are all suggested to help clarify these interesting results. I provide detailed points below:

1. Although the larger trial protocol has been published previously, it is critical to detail when the intervention was started and any other critical details. Please provide details and statistics for average week of gestation at start of intervention for each group
2. Related to this point above, please provide a table that details the baseline characteristics of the study population, split by intervention groups and with statistics/p-values showing differences between intervention groups. Ideally this table will also show details of these characteristics for the larger MISAME-III population to show whether this subset of participants is representative of the larger trial population. A simple table was present in the initial manuscript but not in the revised manuscript. Please re-include this and add details requested above.
3. Also related to point 1 above, if there is large variation between participants in the length of time between intervention initiation and fecal sample collection, it will be important to adjust for that. Can the authors clarify this point and adjust for this if necessary?
4. The top left panel in Fig 1 is only suitable if there are no significant differences in any of these factors between groups, because this image averages the entire population. If there are any differences between groups, I would be inclined to present all of this data in the baseline characteristics table.
5. I am still not entirely happy with the details for how the participants for this sub-study were selected and the cited protocol for BioSpé doesn't appear to elaborate. I understand that this may be a convenience subset (usually limited by funding etc), however it would be important to state if so, and most importantly whether these participants were selected at random from the larger trial. This maybe just needs a little clarification, but as stated above, it would be extremely helpful to provide stats to show whether this sub-population were representative of the larger trial population. Otherwise, this could be a biased sub-population
6. Line 79-80, please clarify if "higher" WAZ and WLZ were significant or not.
7. Line 104 doesn't match the figure 2c that it refers to. It states that prenatal and postnatal Shannon diversity are 4.5 and 3.5 respectively but fig 2c appears to show that median values are all ~4.5 for every group and timepoint. Can you clarify these statistics?
8. The results for increasing Shannon diversity by BEP appear extremely subtle. As suggested above, did you (or could you) adjust for days/weeks since start of the intervention? For example it is still not clear whether the Tri2 samples are taken days or weeks after the start of the intervention, so it is difficult to interpret this significant result.
9. Related to this point above, can you please discuss somewhere how to interpret this result that the intervention would only have an effect prenatally and not postnatally when the intervention is being continued?
10. The distribution of data points in the the PCoA in Fig 2F seem highly unusual (and in the infant panel it is skewed by 3 data outlying data points). I would imagine that this is due to some issue with transformation/filtering/normalization of gene count data. I would suggest to re-check this data before final publication.
11. As above, it will be important to attempt to interpret why gene count data in 2E is only significant at 1 postnatal time point
12. I am not convinced that Fig 3B is very informative. I would suggest to simplify this to simply show total no. shared species between mothers-infants in each treatment group. Given the depth of this sequencing as illustrated in Fig. S1, it is disappointing not to see strain-sharing data, however I appreciate that this is a larger undertaking.
13. Fig 3C needs error bars on each data point. Please also indicate more clearly in the figure that the $p=0.02$ is from a comparison of the first 2 maternal time points
14. Furthermore, I think Fig 3C needs to be adapted to either separate out maternal and infant samples into separate panels, divide the y axis or log transform values (probably the easiest option) so that maternal value can be seen more clearly as the abundances of all maternal values are so low, it is impossible to visualize increases/decreases
15. It is great to see some longitudinal analysis in Fig 3D. However, I am not entirely clear on the stats methods (albeit I am not a statistician). It is still most suitable to assess statistical differences between intervention groups, while accounting for changes over time. It seems to me that the statistical differences have only been assessed over time within treatment groups? For example, I am surprised to see that the *B. fragilis* enrichment noted in the first version of this manuscript has not been sustained in this new analysis. It appears to me from Fig 3D that this is a relatively strong effect of the intervention (in addition to some prevotella species). I would suggest to revisit these statistics to confirm that you are capturing the significant differences between intervention groups, whilst adjusting for the longitudinal effects.
16. Results in Fig 3E are interesting but need a little clarification. Firstly, is this analysis not a form of analysis conducted in Fig 2D and therefore could they not be presented together? Secondly, it is not clear in the mother data which time points are being compared for when analyzing Unifrac distances (is it all timepoints or first and last?).

17. The right panel on Fig 2F is also quite confusing. It seems as if the horizontal lines are error bars? If so, please make them smaller like typical error bars?

18. Network analysis portrayed in Fig 4 adds another element to the analyses but without interpretation it is not that meaningful. Is it possible to analyse which taxa are important or changing in these networks between intervention groups and over time. For example, are any of the taxa from the earlier analyses (e.g. *B. fragilis* or *Prevotella*) important nodes in these networks? Without concrete interpretation of these networks and what they mean with regards to the intervention (i.e. what does increased modularity really mean – is it good/bad?), this analysis may be better suited in supplementary or removed.

19. Overall there are many figures. I would reserve key ones for the main text. I think Fig 7 could go in supplementary. Arguably fig 4 could also go in supplementary, as although the networks looks nice, the figures alone don't tell you much about the results

20. In text describing mediation analysis (lines 256-261) please indicate direction of mediating effects rather than just "significant mediating effect".

21. The mediation analysis results presented in Fig 8B-D is very interesting but difficult to comprehend. As far as I can tell, the blue lines are the direct effect of the intervention on each outcome (gestational age, birth weight etc). The yellow lines are the mediating effect of each taxon. However, if this is correct, then the blue lines should all be exactly the same within each panel? The other alternative is that the blue lines are the direct effect of each taxon on the birth/growth outcome, independently of the intervention. If this alternative is correct, then the results are very difficult to interpret, as almost all results show opposing results for NDE and NIE. For example, in Fig 8C left panel (gestational age), *Ruminococcus D bicirculans* has a very strong positive direct effect on gestational age, increasing it by 5 weeks. However, it has a strong negative mediating effect of the intervention on gestational age, reducing it by ~2.5 weeks. Apart from these effect sizes seeming absolutely huge (are you sure these aren't days?), the results are difficult to comprehend and interpret. I would advise re-checking this analysis and if it remains the same, discussing how to interpret these findings in the discussion.

Reviewer #3

(Remarks to the Author)

The authors have done a decent job revising the earlier manuscript. Most of the concerns raised in the first round of review have been resolved. In particular, the updated analyses as reported in Figures 3 and 4 represent some of the most interesting findings. Given the significance of the dataset generated, the paper is now in very good condition to be published.

Version 2:

Reviewer comments:

Reviewer #1

(Remarks to the Author)

Thank you for your comprehensive responses. I am satisfied that you have addressed all points that were raised. Congrats on the paper!

Reviewer's Comments:

Reviewer #1 (Remarks to the Author)

In this study, Deng and colleagues analyse the gut microbiome of women, and their infants enrolled in a randomized controlled trial of maternal balanced energy-protein supplementation in Burkina-Faso. They report subtle differences in the gut microbiome of supplemented mothers and their infants and report that beneficial effects of BEP supplementation on infant growth are mediated by the gut microbiome. This paper reports an interesting dataset from an underrepresented geographical region with respect to human microbiomes. The results are therefore novel, however greater clarification is required on the analyses and results as outlined below. Major points include clarification of grouped timepoints for analyses and comparisons of timepoints versus groups in addition to expansion of the mediation analyses beyond taxonomic diversity.

- *Abstract: Indicate when BEP was provided i.e. during and after pregnancy?*

Reply: Thank you for this comment, we have now indicated in the abstract that BEP was provided during pregnancy and the first six months of lactation.

- *Line 15-16: This sentence doesn't necessarily support the citation used. This citation looks at 1 month diversity and rapid growth in the context of an obesogenic environment. Therefore, I would disagree with what the authors say that a more mature gut microbiome in the first month may improve infant growth trajectories. Rapid growth isn't always a good thing. I would change this to "influence" infant growth or provide context to say that this study is in a high income setting prone to overweight/obesity*

Reply: We appreciate the feedback and have revised the sentence to use the term "influence" instead of "improve" to avoid implying that rapid growth is inherently beneficial. We agree that context matters, and this change aligns better with the study's setting and scope.

- *In my opinion, the taxonomic tree in Fig 2B doesn't provide any useful information that adds to the results more than what is already presented in Fig 2A. I suggest it is removed/replaced*

Reply: We agree with the reviewer and have removed the taxonomic tree from Figure 2. We have replaced it with a more informative figure that adds new insights (see below).

- *How were participants selected for biospecimen inclusion? Random?*

Reply: Thank you for pointing this out. We have now clarified the selection process for biospecimen inclusion in the Methods section.

Line 1610 (track changes mode):

The BioSpé study was initiated following the completion of recruitment for the MISAME-III trial, when most participants were in their third trimester. To maximize recruitment of participants in

their second trimester, women were prioritized based on gestational age in descending order, ensuring even distribution across all four study groups. Ultimately, 309 women and their infants were enrolled into the BioSpé study, and various biospecimen samples were collected at multiple time points.

- *Figure 2C mother groups all labelled as “postnatal”. Should 2 of these be “prenatal? Also, it is not clear if these include multiple timepoints grouped together? It would be more suitable to separate out each time point as per other figures*

Reply: Thank you for this insightful comment. We have revised Figure 2 to address the concerns raised. Specifically, we have: (1) Replaced panel B with density plots showing the total number of species per sample. (2) Separated out each time points in panel C. (3) Re-colored panels for improved readability and consistency. (4) Updated the group names for clarity by replacing “BEP+IFA” with “Intervention” and “IFA” with “Control”. (5) Throughout the manuscript, we have more clearly emphasized when timepoints were grouped.

Figure 2. Relative abundance, diversity, and gene richness in maternal and infant samples across time points by groups.

(A) Relative abundance of the top 15 most abundant genera (averaged across all samples) aggregated by treatment-groups and sample collection time points. Genera with lower abundance are grouped into the “Other” category.

(B) Density plots showing the total number of microbial species detected per sample in maternal and infant stool samples.

(C) Box-and-jitter plots of Shannon diversity indices grouped by intervention and control at each time point.

(D) Principal Coordinate Analysis (PCoA) ordination plot of microbial species based on weighted UniFrac distance.

(E) Box-and-jitter plots of the gene richness grouped by intervention and control at each time point.

(F) PCoA ordination plot of the gene data based on Bray-Curtis distance.

Interv.: intervention; Contr.: control; Tri2: trimester 2; Tri3: trimester 3; Pn12: postnatal 1-2 months; Pn56: postnatal 5-6 months.

- *Line 91 – “on average higher” – was this significant? Please provide p values etc.*

Reply: Thank you for your comment. We confirm that the difference was statistically significant. We have added the corresponding *p*-value to the revised manuscript to provide more clarity on this point.

Line 808 (track changes mode):

Shannon diversity was significantly higher in prenatal samples compared to postnatal samples ($p = 2.2 \times 10^{-8}$, Wilcoxon rank sum test), with median values of 4.53 bits and 3.52 bits, respectively (Figure 2C).

- *Line 92-93 – please clarify which time points are being compared when stating that there was a significant increase in Shannon diversity. It is not clear from the figure or text between which groups/timepoints there is a significant effect on diversity*

Reply: To clarify, the significant increase in Shannon diversity was observed when comparing the intervention group to the control group using the combined time points of the second and third trimester. We combined these time points due to the limited number of samples from the second trimester and the similarity in Shannon diversity between the second and third trimesters. The timepoint would likely have an effect on diversity, but is unlikely to have an effect on the allocation of the intervention. The combination of timepoints increased the statistical power to detect differences between the groups. We confirm that no significant differences were found when timepoints were analyzed separately. We have revised the manuscript to more clearly indicate this.

Line 812 (track changes mode):

Using the Targeted Maximum Likelihood Estimator (TMLE), no significant average treatment effect of BEP supplementation on Shannon diversity was estimated when analyzing time points separately. Due to the limited number of samples from the second trimester and the similarity in Shannon diversity between the second and third trimester, these time points were combined for further analysis. In this combined analysis, BEP supplementation was estimated to increase

maternal prenatal Shannon diversity index by 0.144 bits (95% CI: 0.009 - 0.280; $p = 0.037$), which corresponds to an estimated increase of 1.16 equally-common species.

- *Line 95-97: why are infant postnatal visit collapsed together? The Shannon diversity at 2 weeks would be expected to be very different to that at 8 weeks.*

Reply: We agree with this comment. We have revised the analysis and corresponding figure (Figure 2) to reflect separate Shannon diversity values for infant samples at each postnatal time point.

- *Figure 2D/E caption colours are wrong*

Reply: Thank you for your observation. We have revised Figure 2 and corrected the colors in the PCoA plots to accurately reflect the intended groups and time points.

- *Clarification on the addition and use of pseudo-counts?*

Reply: We appreciate your request for clarification. We have added the following explanation to the Methods section to clarify the use of pseudo-counts in our analysis:

Line 2079 (track changes mode):

ANCOM-BC2 handles zero-inflation by adding pseudo-counts to the data, which can considerably influence the false-positive and false-negative rates⁵⁶. To mitigate this concern, the ANCOM-BC2 package conducts a sensitivity analysis to evaluate the effect of varying pseudo-counts (ranging from 0.01 to 0.5 in increments of 0.01) on zeros for each taxon. If a taxon is found to be sensitive to pseudo-counts, then it is declared as non-significant taxon⁵⁶. Therefore, only robustly significant taxa were outputted in this analysis.

- *Figure 3: Can you provide a figure with boxplots and individual datapoints for B. fragilis and the Oscillibacter CAG relative abundance? This would be helpful to visualize the difference in abundances between groups at the relevant timepoints*

Reply: We have added the requested boxplots to better visualize the relative abundance of *B. fragilis* and *Oscillibacter CAG*, including individual data points. Additionally, we adjusted the \log_2FC thresholds for visualization from $|\log_2FC| > 1$ to $|\log_2FC| > 0.5$ to capture differential taxa with smaller effects. The revised figure and the following paragraph have been incorporated into the manuscript:

Line 935 (track changes mode):

BEP supplementation is associated with maternal microbial abundances

Differential abundance analyses were conducted separately at each time point, as well as across time points (excluding the second trimester due to limited measurements), with the individual

subject included as a random effect to account for repeated measures. In the individual time point analyses (Figure 5A), 29 taxa in maternal samples were significantly associated with the intervention, 16 of which had a log₂ fold change (log₂FC) greater than 0.5. Among these, two species exhibited larger biological effect sizes: depletion of an *Oscillospiraceae* species [CAG-103 sp. 000432375; $p = 2.48 \times 10^{-4}$; log₂FC = -1.12] in the second trimester and enrichment of *B. fragilis* ($p = 3.26 \times 10^{-8}$; log₂FC = 1.31) at 5-6 months postpartum. In contrast, no robustly significant associations were found between BEP supplementation and the microbial abundances in infant stool samples at either time point. Analogously, in the combined time point analysis, no significant associations were found between the intervention and microbial abundance in either maternal or infant samples (Figure 5B). The full result table of differential abundance analysis is available in Table S1.

Figure 5. Differential taxa identified by ANCOM-BC2 in maternal and infant samples across time points.

(A) Volcano plots showing differential abundant taxa associated with the intervention at each time point for both maternal and infant samples.

(B) Volcano plots depicting taxa differentially abundant between the intervention and control groups when time points are combined (excluding the second trimester due to limited measurements). The x-axis represents log₂ fold change (log₂FC) in abundance, and each point represents a taxon. Horizontal dashed lines indicate the thresholds of statistical significance (i.e., Benjamini-Hochberg adjusted *P*-values < 0.05), and vertical dashed line represent |log₂FC| > 0.5. Colored dots indicate taxa significantly enriched (blue) and depleted (red) in the intervention group compared to the control group. Results were confirmed through sensitivity analysis for pseudo-counts. A full table of ANCOM-BC2 outputs can be found in Table S1.

(C) Boxplots displaying the relative abundance of the 16 taxa with |log₂FC| > 0.5 that were significantly different between the intervention and control groups.

Interv.: intervention; Contr.: control; Tri2: trimester 2; Tri3: trimester 3; Pn12: postnatal 1-2 months; Pn56: postnatal 5-6 months.

- *Overall, the separation of “prenatal” and “postnatal” results makes things a little confusing, as it is clearly indicated in the methods that only the combined prenatal+postnatal intervention groups were included in this study*

Reply: Thank you for highlighting this potential source of confusion. To clarify, the statistical analyses were indeed conducted separately for each time point to assess differences between the intervention and control groups. We have revised the Results section to specify the comparisons at each time point more clearly. Additionally, to enhance readability, we have updated the Methods section and the main text to consistently refer to the groups as the "intervention group" (formerly BEP+IFA group) and the "control group" (formerly IFA group).

Line 1622 (track changes mode):

Fecal samples were collected from mother-infant dyads across all four trial arms, but only two groups of samples (*n* = 152) were sequenced: 71 pairs from the intervention group (receiving both pre- and postnatal BEP and IFA supplementation) and 81 pairs from the control group (receiving both pre- and postnatal IFA tablets only).

- *In the mediation analysis, more detail is needed on the direction of effects in lines 115-118. It seems from figure 2 that BEP, if anything, decreases infant diversity, albeit not significantly, suggesting the mediation effect is via reducing Shannon diversity. I think it would be helpful to expand figure S5 but to include the NIE and NDE values on each arrow/interaction to show the strength and direction of each effect. This figure could be put in the main text as it is central to the story.*

Reply:

Thank you for your valuable feedback. We re-analyzed our data by separating the postnatal time points. The causal mediation analysis indicated that BEP supplementation had a direct effect on

gestational age and growth outcomes, but these effects were not mediated by overall diversity. To provide more detail, we expanded our mediation analysis to include individual taxonomic data. The analysis showed that specific microbial taxa played a mediating role, and we could attribute this effect to some specific species. As suggested, we have expanded the Figure S5 by including the NIE and NDE values on each arrow to show the strength and direction of these effect. We have moved this figure to the main text (now Figure 8) and stated the NDE values in the main text (see below the reply to the second Major point).

- *Please also clarify the “timepoint” in table S2 to confirm if the mediation is with combined timepoints.*

Reply:

The timepoints under consideration in the mediation analyses were indeed not clear. We have expanded Table S2 (now Table S3) to include this information.

- *Major point: Due to the major limitation of no true baseline sample, it is important to indicate in all results what the comparison group is when indicating a significant change in diversity/abundance etc. For example, in lines 131-147 please indicate if the “enrichment” of genes in BEP-supplemented mothers is in comparison to earlier timepoints in the same mothers or in comparison to non-supplemented mothers at the same time point.*

Reply:

This is indeed an important point. In the specified section (prior: lines 131-147, now lines 961-1211 under track changes mode), the “enrichment” of genes in BEP-supplemented mothers refers to a comparison with the non-supplemented control group at the same time point, rather than earlier time points within the same mothers. We have revised the manuscript to clearly and consistently indicate the comparison group in all relevant sections, ensuring that all comparisons are clearly specified as differences between the intervention and control groups. This revision should help improve the clarity and interpretability of our findings. Furthermore, we conducted a more detailed analysis of the changes in microbial composition across the various time points. However, it remains true that establishing a true baseline for comparison is not feasible within the context of our study. We have acknowledged this in the limitations in the Discussion section.

- *Line 157-158: The “striking disparity” between mothers and their infants wouldn’t usually be considered a “novel finding” as this has been reported in every dataset of maternal-infant microbiomes around the world.*

Reply: Thank you for pointing out this issue. We agree that this is not a novel finding, and we have removed this point from the “novel finding” statement.

- *Major point: Mediation analysis is currently only for taxonomic diversity and not for other things such as individual pathways/taxa or PC1/PC2 score. Can you expand the mediation analysis to include some of these things?*

Reply:

Thank you for your valuable suggestion. We have now included a comprehensive mediation analysis with multivariate microbial taxa as mediators. These taxa can however have interconnected effects on both each other and on the assessed outcomes, opening up paths of possible intermediate confounding. The methods used here thus account for this possibility and estimate interventional (in)direct effects rather than natural effects. Although these can be slightly more complex to interpret, they can provide insight into the causal relationships behind the biological mechanisms set in action by BEP supplementation, which was likely the reviewer's intent with this suggestion.

We have opted not to incorporate a mediation analysis on beta diversity coordinates, as these would violate the 'no-multiple versions of treatment' assumption that is core to causal inference. Given the unknown meaning of the axes resulting from ordination, interpreting the indirect effects estimated by such a mediation analysis would likely also be highly challenging.

As for the mediation analysis on functional data, pathways abundances as mediators could be highly interesting. However, genes can be part of multiple pathways at the same time, making them their own intermediate confounders. We are not aware of any methods that tackle this issue and believe the current state-of-the-art to be insufficient to perform a mathematically correct functional mediation analysis using pathways as mediators. Instead of this direct pathway mediation, we conducted a multivariate mediation analysis on the top 500 individual genes and performed a gene set enrichment analysis on the indirect effect estimates of this analysis to scope out any coordinated effects. We note that the ensuing gene set enrichment analysis did not reveal significant microbial pathways.

These expanded analyses provide a more comprehensive understanding of how the BEP supplementation may influence birth outcomes and infant growth through specific microbial features. We have updated the manuscript to incorporate these new analyses and have included the relevant results in the main text and Figure 8.

Line 1213 (track changes mode):

Microbiome modulation mediates birth outcomes and infant growth in response to BEP supplementation

We conducted causal mediation analyses to elucidate the role of the microbiome (through changes in either Shannon diversity or individual taxon abundance) in the downstream effects of BEP supplementation on birth outcomes (i.e., SGA, gestational age, birth weight and length) and infant growth metrics (i.e., LAZ, WAZ and WLZ) at 3 and 6 months of age.

The mediation analyses using the Shannon diversity of the microbial communities as a mediator revealed significant natural direct effects of the intervention on gestational age of the infants at birth (0.56 weeks; 95% CI: 0.05 – 1.08 weeks), as well as on WAZ at 3 months (0.43; 95% CI: 0.03 – 0.83) and at 6 months (0.43; 95% CI: 0.04 – 0.82) and on the odds of stunting at 6 months (-11.6%; 95% CI: -21.5 – -1.6%) (Figure 8A). However, no significant natural indirect effects were observed, indicating that these effects were not mediated by shifts in microbial diversity. Therefore, we conducted a multivariate mediation analysis using individual taxon abundances. In the second trimester, specific taxa exhibited a significant mediating effect on birth outcomes. The abundances observed after the intervention of species from order Oscillospirales (UMGS911 sp900545935), *Lachnospira* (sp900316325), *Faecalibacterium prausnitzii*, and *Butyricoccus A* (sp002395695) had significantly positive effects on gestational age, birth weight, and length. Conversely, the intervened abundances of the order Bacteroidales (RC9 sp900546925 and RC9 sp000434935), *Oscillospiraceae* (CAG-83 sp900547745), *Phascolarctobacterium succinatutens*, *Prevotella* sp900553155, *Gastranaerophilaceae* (UMGS1585), and *Prevotellamassilia* sp900543155 had negative effects on anthropometric outcomes (Figure 8B).

In the third trimester, *Ruminococcus bicirculans* and species from the order Christensenellales (CAG-1435), Bacteroidales (RC9 sp900546925), Oscillospirales (UMGS1002), and *Lachnospiraceae* (CAG-632 sp000431515) were associated with decreases in gestational age, birth weight, and length in mothers receiving BEP supplementation. Meanwhile, *Oscillospiraceae* CAG-110 sp900556935 showed a positive mediating effect gestational age, birth weight, and length in mothers receiving BEP supplementation, without affecting the odds of SGA (Figure 8C). At 1 to 2 months of age, the abundances of *Bifidobacterium breve* and *Streptococcus* sp000187445 had significant mediating effects on both WAZ [*B. breve*: -0.13 (-0.24 – -0.01); *S. sp000187445*: 0.07 (0.01 – 0.12)] and WLZ [*B. breve*: -0.15 (-0.28 – -0.02); *S. sp000187445*: 0.08 (0.02 – 0.14)] at six months (Figure 8D). *Bifidobacterium infantis* also displayed significant mediating effects on LAZ and WAZ at 3 and 6 months, though with negligible effect size (Table S3).

We also conducted causal mediation analysis on gene-level data by selecting the top 500 genes. While specific genes exhibited significant mediating effects on birth outcomes and infant growth, subsequent gene set enrichment analysis did not reveal distinct, significant microbial pathways. The complete results from the causal mediation analyses are available in Table S3.

Figure 8: Mediating effect of the microbiome on birth outcomes and infant anthropometry.

(A) Estimates for the Natural Indirect Effect (NIE) and the Natural Direct Effect (NDE) of the microbiome diversity on gestational age at birth and on length for age at six months. Results are superimposed on the causal directed acyclic graphs used in these analyses. The model with maternal Shannon diversity as mediator included maternal stratum data as covariates (level of education, marital status, language, religion, ethnicity, number of jobs, socio-economic status, dietary diversity score, and health center), as well as maternal anthropometry taken at inclusion into the trial (age, height, BMI and hemoglobin level). The model with infant microbiome diversity as a mediator also included the age at which the infant stopped exclusively breastfeeding as a covariate. Infant microbiome diversities were measured at 1 to 2 months of age. (B) Estimates for the interventional direct and indirect effects of each individual species' abundance in the maternal microbiome in the second trimester, (C) the third trimester, and (D) in the infant

microbiome at 1-2 months of age. The complete results of these mediation analyses are available in Table S3.

- *Supplementary tables need a key to explain what each column header means.*

Reply:

Thank you for your comments, we acknowledge that these headers are not necessarily meaningful on their own. We have now added a sheet with column descriptions to Tables S1, S2, and S3.

- *It may be useful to check metagenomic diversity (gene richness), its changes with supplementation and its mediating effect. Similarly, it would be beneficial to look at functional B-diversity as most of the effects were in functional data.*

Reply: This is a valuable suggestion. In response, we have expanded our analysis to include gene richness and β -diversity, comparing these metrics between the intervention and control groups.

The results have been incorporated into Figure 2, and we have added a corresponding paragraph to the Results section to describe these findings. As mentioned above, we performed a mediation analysis on gene counts, followed by a gene set enrichment on these estimates, but we did not include a mediation analysis specifically on gene richness. We report the gene richness and beta-diversity in the main text and Figure 2F.

Line 824 (track changes mode):

Similarly, microbial gene richness was lower in infant samples compared to maternal samples, with a median observed gene richness of 27,558 in infants and 36,340 in mothers (Figure 2E). TMLE analysis suggested that BEP supplementation increased gene counts in maternal samples at 1-2 months postpartum by 2,007 counts (95% CI: 95 – 3920; $p = 0.040$). No significant associations between BEP supplementation and gene counts were found at the other time points for maternal or infant samples. Although beta-diversity of gene data showed diversification of the infant metagenome over time ($p = 0.007$), no significant differences were attributed to the intervention ($p = 0.631$). Similarly, no significant differences in maternal genome beta-diversity were observed related to either the time point ($p = 0.111$) or the intervention ($p = 0.437$) (Figure 2F).

Reviewer #2 (Remarks to the Author):

Dear Authors,

Thank you for providing a well written manuscript in a population which is understudies. I will provide general feedback on the four main outcomes outlined in the discussion followed by more specific feedback.

The main findings are:

1- Striking disparity in the gut microbiome composition between mothers and their infants.

This is not a novel finding, multiple papers have found this and the infant microbiome is distantly different from the adult gut microbiome. This should be acknowledged and the finding should not be featured as novel.

Reply: Thank you for pointing out this issue. We agreed that this is not a novel finding, and we have removed this point from the “novel finding” statement.

2- BEP supplementation affected specific facets of the maternal gut microbiome, increase in prenatal microbial diversity, with a decrease in the abundance of an Oscillospiraceae species during pregnancy and an increase of Bacteroides fragilis at 5-6 months postpartum.

BEP supplementation is a significant addition to the diet (72 g per day) and it would therefore be expected to result in changes to the maternal gut microbiome. Because of the population group the effect of the supplement is likely to have a greater effect in this population compared to for example a Western population. The change is interesting and maybe novel for this population group and with this intervention, but not unexpected. A diet supplementation should be seen in relation to the habitual diet of the individual. However, no diet information is given in the manuscript it's therefore not possible for the reader to evaluate the impact of the supplement on diet intake. It would be interesting to know how much the supplement increase the protein intake as that will affect the carbohydrate/protein ratio which is very important for changing the gut microbiome.

Reply: Thank you for your insightful comments. We agree that understanding the habitual diet is crucial for evaluating the impact of BEP supplementation on the gut microbiome. In the original manuscript, we briefly described the dietary pattern in the Methods section. Based on your suggestion, we have moved this information to the Introduction section and elaborated on it to provide a clearer context regarding the habitual diet of participants.

Line 299 (track changes mode):

Prior studies have shown that the diet of women who participated in MISAME-III trial was nondiverse, with only 45% of participants reaching Minimum Dietary Diversity for Women (i.e., ≥ 5

food groups)²⁶. The base diet is mainly cereal-based, with leafy vegetables as a supplement, contributing to half the total energy intake; other nutritious food groups, such as fruit, dairy, eggs, fish, and meat contribute a very small amount to the total energy intake^{27,28}. BEP supplementation significantly improved both energy intake and micronutrient adequacy without displacing other nutrient sources. On average, women in the intervention group consumed an additional 15.2g of protein, 37.5g of carbohydrate and 25.6g of fat compared to the control group on a daily basis²⁸.

3- Mediation analysis revealed that BEP supplementation drives infant growth through microbiome diversity.

The evidence provided doesn't convince me that it's the microbiome diversity that is driving the growth of the infant as it's likely that a mother having additional food (BEP) produces more milk and therefore the infant thrives better. Therefore, the change in microbiome diversity is a secondary effect and not the driver of the effect. It should also be kept in mind that no difference in beta-diversity was found between supplementation and control group.

Reply: Thank you for your insightful comments. Our mediation analysis aimed to assess the specific role of the microbiome in this process. We understand your concern about whether microbiome diversity directly drives infant growth. After reanalyzing the data and separating postnatal time points, we found that the Natural Indirect Effect (NIE) through microbiome diversity was no longer significant. Instead, our expanded analysis revealed that specific microbial taxa, rather than overall diversity, played a more prominent role in mediating the effects of BEP supplementation on infant growth. We would like to provide a clearer understanding of the NIE and Natural Direct Effect (NDE) better. The mediation analysis, (Figure S5 in the original manuscript), now presented as Figure 8 in the main text, specifically estimates the NIE, which isolates the effect of BEP supplementation through microbiome on the outcomes. This means that any potentially post-treatment effect caused by BEP, such as a higher volume of milk, would still impact the outcome (e.g., better infant growth), but those effects would be captured in the estimation of the NDE. By isolating the NIE, we quantitatively determine whether microbial diversity itself plays a role in this process, regardless of any other causal agents. Additionally, we emphasize that mediators, such as microbial diversity or taxa in this case, are not ‘drivers’ of the process but are rather intermediaries through which the effect of the exposure (BEP) can amplify or diminish. We have revised the whole mediation analysis section to reflect these findings.

Line 917 (track changes mode):

Microbiome modulation mediates birth outcomes and infant growth in response to BEP supplementation

We conducted causal mediation analyses to elucidate the role of the microbiome (through changes in either Shannon diversity or individual taxon abundance) in the downstream effects of BEP supplementation on birth outcomes (i.e., SGA, gestational age, birth weight and length) and infant growth metrics (i.e., LAZ, WAZ and WLZ) at 3 and 6 months of age.

The mediation analyses using the Shannon diversity of the microbial communities as a mediator revealed significant natural direct effects of the intervention on gestational age of the infants at birth (0.56 weeks; 95% CI: 0.05 – 1.08 weeks), as well as on WAZ at 3 months (0.43; 95% CI: 0.03 – 0.83) and at 6 months (0.43; 95% CI: 0.04 – 0.82) and on the odds of stunting at 6 months (-11.6%; 95% CI: -21.5 – -1.6%) (Figure 8A). However, no significant natural indirect effects were observed, indicating that these effects were not mediated by shifts in microbial diversity. Therefore, we conducted a multivariate mediation analysis using individual taxon abundances.

In the second trimester, specific taxa exhibited a significant mediating effect on birth outcomes. The abundances observed after the intervention of species from order Oscillospirales (UMGS911 sp900545935), *Lachnospira* (sp900316325), *Faecalibacterium prausnitzii*, and *Butyricicoccus* A (sp002395695) had significantly positive effects on gestational age, birth weight, and length. Conversely, the intervened abundances of the order Bacteroidales (RC9 sp900546925 and RC9 sp000434935), *Oscillospiraceae* (CAG-83 sp900547745), *Phascolarctobacterium succinatutens*, *Prevotella* sp900553155, *Gastranaerophilaceae* (UMGS1585), and *Prevotellamassilia* sp900543155 had negative effects on anthropometric outcomes (Figure 8B).

In the third trimester, *Ruminococcus bicirculans* and species from the order Christensenellales (CAG-1435), Bacteroidales (RC9 sp900546925), Oscillospirales (UMGS1002), and *Lachnospiraceae* (CAG-632 sp000431515) were associated with decreases in gestational age, birth weight, and length in mothers receiving BEP supplementation. Meanwhile, *Oscillospiraceae* CAG-110 sp900556935 showed a positive mediating effect gestational age, birth weight, and length in mothers receiving BEP supplementation, without affecting the odds of SGA (Figure 8C).

At 1 to 2 months of age, the abundances of *Bifidobacterium breve* and *Streptococcus* sp000187445 had significant mediating effects on both WAZ [*B. breve*: -0.13 (-0.24 – -0.01); *S. sp000187445*: 0.07 (0.01 – 0.12)] and WLZ [*B. breve*: -0.15 (-0.28 – -0.02); *S. sp000187445*: 0.08 (0.02 – 0.14)] at six months (Figure 8D). *Bifidobacterium infantis* also displayed significant mediating effects on LAZ and WAZ at 3 and 6 months, though with negligible effect size (Table S3).

We also conducted causal mediation analysis on gene-level data by selecting the top 500 genes. While specific genes exhibited significant mediating effects on birth outcomes and infant growth, subsequent gene set enrichment analysis did not reveal distinct, significant microbial pathways. The complete results from the causal mediation analyses are available in Table S3.

Figure 8: Mediating effect of the microbiome on birth outcomes and infant anthropometry.

(A) Estimates for the Natural Indirect Effect (NIE) and the Natural Direct Effect (NDE) of the microbiome diversity on gestational age at birth and on length for age at six months. Results are superimposed on the causal directed acyclic graphs used in these analyses. The model with maternal Shannon diversity as mediator included maternal stratum data as covariates (level of education, marital status, language, religion, ethnicity, number of jobs, socio-economic status, dietary diversity score, and health center), as well as maternal anthropometry taken at inclusion into the trial (age, height, BMI and hemoglobin level). The model with infant microbiome diversity as a mediator also included the age at which the infant stopped exclusively breastfeeding as a covariate. Infant microbiome diversities were measured at 1 to 2 months of age. (B) Estimates for the interventional direct and indirect effects of each individual species' abundance in the maternal microbiome in the

second trimester, (C) the third trimester, and (D) in the infant microbiome at 1-2 months of age. The complete results of these mediation analyses are available in Table S3.

*4- Multiple microbial pathways associated with BEP supplementation, of which, the PTS genes for carbohydrate uptake were enriched in infant samples whose mothers were BEP-supplemented
Could this be related to the adaptation of the microbiome to changes in the infant gut related to better maternal nutrition affecting the infant through breastfeeding?*

I did notice that the initial maternal sampling time had a reduced number of samples due to the sample collection starting after the main study. However, for the remaining timepoint it would have been interesting to know how many mother-infant dyads had provided samples at all time points. This could have allowed for a repeated measure analysis which could remove some of the variability between timepoints.

Reply: Thank you for your comments. We agree that the changes observed in the infant gut microbiome may indeed be related to improved maternal nutrition, potentially affecting the infant gut through breastfeeding. However, we do not have solid evidence directly link maternal BEP supplementation with changes in breast milk composition that could explain these microbiome adaptations in infants. Ongoing analyses of breast milk composition aim to confirm this mechanism. This relationship has been included in the Discussion section.

Line 1050 (track changes mode):

Functionally, maternal BEP supplementation was associated with enriched carbohydrate metabolism pathways in the infant gut, such as PTS and starch metabolism, potentially influenced by changes in breast milk composition.

We also appreciate your suggestion regarding repeated measure analysis. Apart from the inclusion time point, 105 mother-infant dyads ($n = 50$ in the control group and $n = 55$ in the intervention group) provided samples at all other time points. We performed a differential abundance analysis based on these repeated measures to assess the relative abundance of each taxon between the intervention and control groups. However, no significant differences were found. This has been described in the revised Results section. Additionally, we conducted a mixed effect model to track changes in the relative abundance of the top ten most abundant species over time. These results, along with the corresponding figures, have been updated in the manuscript:

Line 615 (track changes mode):

Overall, except for the inclusion time point, 105 mother-infant dyads ($n = 55$ in the intervention group and $n = 50$ in the control group) provided samples at all other time points.

Line 896 (track changes mode):

Figure 3D presents the top ten taxa with a minimum abundance of 1% and presence in at least five samples across all time points, categorized by intervention groups for both maternal and infant samples. A mixed-effects linear regression model, with subject identifier as a random effect, was used to compare changes in relative abundance over time between groups. Among the top ten taxa in maternal samples, two exhibited significantly greater increases in the intervention group compared to the control group: *Prevotella* sp002299635 (from the third trimester to 5-6 months postpartum; $p = 0.04$) and Bacteroidales RC9 sp000433355 (from the third trimester to 1-2 months postpartum; $p = 0.02$).

Our analysis revealed that the infant gut microbiome in the intervention group changed more rapidly over time compared to the control group ($p = 0.04$), as evidenced by mixed-effect linear regression models with intra-subject distance (calculated using weighted UniFrac) as the dependent variable and days interval as the independent variable (Figure 3E). In maternal samples, the intervention demonstrated a heterogenous effect on pre- and postpartum microbiome compositions (Figure 3F).

Line 928 (track changes mode):

Differential abundance analyses were conducted separately at each time point, as well as across time points (excluding the second trimester due to limited measurements), with the individual subject included as a random effect to account for repeated measures. In the individual time point analyses (Figure 5A), 29 taxa in maternal samples were significantly associated with the intervention, 16 of which had a log₂ fold change (log₂FC) greater than 0.5. Among these, two species exhibited larger biological effect sizes: depletion of an *Oscillospiraceae* species [CAG-103 sp. 000432375; $p = 2.48 \times 10^{-4}$; log₂FC = -1.12] in the second trimester and enrichment of *B. fragilis* ($p = 3.26 \times 10^{-8}$; log₂FC = 1.31) at 5-6 months postpartum. In contrast, no robustly significant associations were found between BEP supplementation and the microbial abundances in infant stool samples at either time point. Analogously, in the combined time point analysis, no significant associations were found between the intervention and microbial abundance in either maternal or infant samples (Figure 5B). The full result table of differential abundance analysis is available in Table S1.

Figure 3. Temporal stability of maternal and infant gut microbiomes across time points by groups.

(A) Bar plots of taxa recurrence rate within each phylum.

(B) UpSet plot showing the number of shared species across different time points and mother-infant pairs.

(C, D) Line plots displaying the relative abundance of (C) species shared between mothers and infants across time points and (D) the top ten most abundant species in both mother and infant samples. The species shown have a minimum abundance of 1% and are present in at least five dyads samples across the time points (excluding the second trimester due to limited measurements).

(E) Linear regression between dissimilarity (beta diversity) against collection date interval, comparing intra-subject microbiome stability across groups.

(F) Shifts in microbiomes over time, assessed by intra-subject distance using weighted UniFrac. *P*-values were adjusted using the Benjamini-Hochberg correction ($p < 0.05$).

Figure 5. Differential taxa identified by ANCOM-BC2 in maternal and infant samples across time points.

(A) Volcano plots showing differential abundant taxa associated with the intervention at each time point for both maternal and infant samples.

(B) Volcano plots depicting taxa differentially abundant between the intervention and control groups when time points are combined (excluding the second trimester due to limited measurements). The x-axis represents log₂ fold change (log₂FC) in abundance, and each point represents a taxon. Horizontal dashed lines indicate the thresholds of statistical significance (i.e., Benjamini-Hochberg adjusted *P*-values < 0.05), and vertical dashed line represent $|\log_2FC| > 0.5$. Colored dots indicate taxa significantly enriched (blue) and depleted (red) in the intervention group compared to the control

group. Results were confirmed through sensitivity analysis for pseudo-counts. A full table of ANCOM-BC2 outputs can be found in Table S1.

(C) Boxplots displaying the relative abundance of the 16 taxa with $|\log_2FC| > 0.5$ that were significantly different between the intervention and control groups.

Interv.: intervention; Contr.: control; Tri2: trimester 2; Tri3: trimester 3; Pn12: postnatal 1-2 months; Pn56: postnatal 5-6 months.

L 98 - States that infant had between 2 and 43 species. I would question if a sample with only 2 identified species is a true representation of the microbiome.

Reply: We appreciate your comment regarding the sample with only two identified species and its representation of the microbiome. We employed deep sequencing techniques and genome base analysis, which maximize the detection sensitivity, even for low-abundance species, thereby providing a robust representation of the microbiome. Additionally, we used a reference database specifically tailored for low- and middle-income countries, which considers the unique microbiome characteristics from these regions. In the revised Figure 2B, we present a density plot that illustrates the distribution of identified species across all samples, and we note that infants with only two identified species are not outliers. Upon review, we found that these samples are dominant by *Bifidobacterium longum*, constituting 98% of the microbial abundance, which aligns with expected patterns in exclusively breastfed infants (PMID: 38087708). The raw read counts for these species are consistent with other samples, suggesting that the sequencing depth and quality were adequate. Thus, these samples are valid representations of the infant microbiome within the context of our study.

Figure 2. Relative abundance, diversity, and gene richness in maternal and infant samples across time points by groups.

(A) Relative abundance of the top 15 most abundant genera (averaged across all samples) aggregated by treatment-groups and sample collection time points. Genera with lower abundance are grouped into the “Other” category.

(B) Density plots showing the total number of microbial species detected per sample in maternal and infant stool samples.

(C) Box-and-jitter plots of Shannon diversity indices grouped by intervention and control at each time point.

(D) Principal Coordinate Analysis (PCoA) ordination plot of microbial species based on weighted UniFrac distance.

(E) Box-and-jitter plots of the gene richness grouped by intervention and control at each time point.

(F) PCoA ordination plot of the gene data based on Bray-Curtis distance.

Interv.: intervention; Contr.: control; Tri2: trimester 2; Tri3: trimester 3; Pn12: postnatal 1-2 months; Pn56: postnatal 5-6 months.

Overall, I find the research interesting but the outcome is as expected.

Reply: Thank you for your comments and valuable feedback!

Reviewer #3 (Remarks to the Author):

Deng et al. present an analysis of data collected from mother-infant dyads in a randomized controlled trial conducted in rural Burkina Faso. The trial aimed to study the effects of balanced energy-protein (BEP) supplementation on the gut microbiome. This data is unique, primarily due to metagenomic sequencing and a relatively large sample size from an underrepresented population. While the data collected are impressive and certainly serve as a valuable resource, there are multiple shortcomings in the analyses conducted. In particular, the treatment of the statistical analysis is unfocused and includes sporadic information without clear direction. A more effective approach would have been concentrating on fewer key results, analyzing and presenting in greater depth. As a result, the overall impact is diluted. My specific comments are as follows:

1. Although the study had unique access to mother-infant dyads at multiple time points, none of the analyses considered jointly modeling the mother-child pair as a single analytical unit, as is common practice in maternal-fetal human genetics studies. This oversight represents a significant missed opportunity to leverage the potential of the data fully.

Reply: Thank you for your insightful feedback regarding the potential for jointly modeling the mother-infant dyads as a single unit. We appreciate your suggestion and the opportunity to clarify our approach. For this analysis, our primary objective was to assess the effect of BEP supplementation on maternal and infant gut microbiomes separately, as well as to explore their mediating effects on birth outcomes and infant growth. Given the distinct differences between maternal and infant gut microbiomes—shaped by both biological and environmental factors—combining these into a single model would have potentially obscured any the specific effects of the intervention that we aimed to investigate.

In the revised manuscript, we have explored the connection between maternal and infant microbiomes through analyses that visualize the numbers of shared species and the relative abundance of the top ten shared species in mother-infant pairs, comparing their changes across time between the intervention and the control groups. These analyses allow us to understand the microbiome connections and dynamics within dyads. These results, along with the corresponding figures, have been updated in the manuscript (Line 875 and Figure 3, see below).

2. The availability of longitudinal samples at both pre- and post-intervention time points provided a unique opportunity for dynamic analysis. However, no assessment of temporal variability in taxonomic and functional composition was described. As a result, the study failed to capitalize on the

time-varying aspect of gut microbiome changes, which diminishes the depth of the findings and overlooks potential insights into the developmental trajectories of the microbiome.

Reply: Thank you for your thoughtful feedback on the potential for dynamic analysis of temporal variability in taxonomic and functional composition. We appreciate the value of leveraging longitudinal data to uncover developmental trajectories of the microbiome.

We would like to clarify that, unfortunately, this study does not include a true pre-intervention time point. The study was initiated after the recruitment for the MISAME-III trial had been completed, at which point most participants were already in their third trimester of pregnancy. As a result, the number of samples collected at the time of inclusion (i.e., during the second trimester) was significantly fewer (around 20 per group) compared to subsequent time points (around 70 per group). Additionally, by the time of inclusion, the women had already been receiving BEP supplementation for 2-3 weeks, limiting our ability to assess the prenatal gut microbiome prior to the onset of supplementation.

Despite this limitation, we recognize the importance of capturing temporal variability in the microbiome and aimed to address this through longitudinal analysis. We assess changes in relative abundance over time, focusing on the shared species between mother-infant pairs and the top ten most abundant species in maternal or infant samples. We excluded the second trimester for maternal samples due to the limited number of samples available at that stage. In addition, we performed linear regression of intra-subject distance over time, which revealed that the infant microbiome in the intervention group changed more rapidly compared to the control group. These analyses provide insights into the temporal variability in the microbiome, despite the absence of a true pre-intervention baseline. These analyses have been included in the revised manuscript (Line 875 and Figure 3).

Line 875 (track changes mode):

BEP supplementation is associated with microbial temporal stability

In maternal samples, the majority of species were consistently present across all time points in both the intervention and control groups. *Firmicutes* A, the most abundant phylum, constituted up to 75% of the microbial community at all time points. In contrast, infant samples exhibited less species recurrence, with fewer than half of the species consistently present across all time points (Figure 3A). Across all samples, 2,654 unique species were detected in at least one sample, of which 2,493 were exclusive to the maternal population, 396 were unique to the infant population, and 235 species were shared between mothers and infants. Among the shared microbial genomes, 205 were already observed in the mothers during prenatal time points. Approximately 50% of the microbial species found in the infant gut were also detected in at least one maternal sample, and this fraction remained relatively stable over time, with 36% in the second trimester, 45% in the third trimester, 44% at 1-2 months postpartum, and 40% at 5-6 months postpartum.

Interestingly, a higher number of shared species was observed between mothers and infants in the control group compared to the intervention group ($p = 0.019$, chi-square test). Specifically, 165 out of 281 species were shared in the control group, whereas 142 out of 292 species were shared in the intervention group (Figure 3B). Of the nine shared species present in at least five samples at each time point, the relative abundance of these shared species was notably lower in the mothers compared with their infants (Figure 3C). Additionally, a strain of *Bacteroides fragilis* exhibited significantly greater increases in relative abundance in the intervention group compared to the control group from the third trimester to 1-2 months postpartum ($p = 0.02$).

Figure 3D presents the top ten taxa with a minimum abundance of 1% and presence in at least five samples across all time points, categorized by intervention groups for both maternal and infant samples. A mixed-effects linear regression model, with subject identifier as a random effect, was used to compare changes in relative abundance over time between groups. Among the top ten taxa in maternal samples, two exhibited significantly greater increases in the intervention group compared to the control group: *Prevotella* sp002299635 (from the third trimester to 5-6 months postpartum; $p = 0.04$) and Bacteroidales RC9 sp000433355 (from the third trimester to 1-2 months postpartum; $p = 0.02$).

Our analysis revealed that the infant gut microbiome in the intervention group changed more rapidly over time compared to the control group ($p = 0.04$), as evidenced by mixed-effect linear regression models with intra-subject distance (calculated using weighted UniFrac) as the dependent variable and days interval as the independent variable (Figure 3E). In maternal samples, the intervention demonstrated a heterogeneous effect on pre- and postpartum microbiome compositions (Figure 3F).

Figure 3. Temporal stability of maternal and infant gut microbiomes across time points by groups.

(A) Bar plots of taxa recurrence rate within each phylum.

(B) UpSet plot showing the number of shared species across different time points and mother-infant pairs.

(C, D) Line plots displaying the relative abundance of (C) species shared between mothers and infants across time points and (D) the top ten abundant species in both mother and infant samples. The species shown have a minimum abundance of 1% and are present in at least five samples across the time points (excluding the second trimester due to limited measurements).

(E) Linear regression between dissimilarity (beta diversity) against collection date interval, comparing intra-subject microbiome stability across groups.

(F) Shifts in microbiomes over time, assessed by intra-subject distance using weighted UniFrac. *P*-values were adjusted using the Benjamini-Hochberg correction ($p < 0.05$).

3. The major conclusions of the paper focus solely on microbiome diversity. While diversity is a useful metric, it is only one aspect of understanding an ecosystem's complexity. Current literature indicates that microbiome diversity alone does not provide a complete picture and must be considered alongside other factors such as stability, structure, and function. This is particularly relevant, as comparisons of diversity between two clinical groups often yield contradictory results. Therefore, a broader analytical approach, based on specific microbiome features (beyond what is conducted in the manuscript), is necessary to comprehensively support the conclusions.

Reply: Thank you for this important point. We agree that microbiome diversity alone provides only a partial understanding of ecosystem complexity. In response, except for the temporal stability analysis (see response to comment 2), we have broadened our analysis beyond diversity to include microbial community structure and stability by performing a co-occurrence network analysis. This approach allowed us to explore the microbial interactions, compartmentalization (modularity), and connectivity (density, clustering coefficients) between taxa in different groups. The results of this analysis, now included in Figure 4, provide deeper insights into how BEP supplementation alters the gut microbiome beyond diversity metrics alone. These analyses have been included in the revised manuscript (Line 912 and Figure 4).

Line 912 (track changes mode):

BEP supplementation alters the maternal gut microbiome's network structure over time

The co-occurrence network analysis revealed distinct shifts in microbial community structure between the intervention and control groups in maternal samples, particularly in relation to network modularity (Figure 4). During prenatal time points, the intervention group exhibited a higher number of nodes and edges compared to the control group. In the second trimester, Louvain community detection identified more communities in the intervention group, alongside significantly lower modularity ($p = 0.016$), suggesting less compartmentalization and more interconnectivity within the microbial network. In the third trimester, although network density and clustering coefficients were similar between groups, the network diameter in the intervention group was significantly smaller ($p = 0.021$), reflecting more compact interactions between taxa. During postnatal time points, the network structure shifted, with the intervention group displaying fewer nodes and edges but significantly greater modularity at both 1-2 months postpartum ($p = 0.002$) and 5-6 months postpartum ($p < 0.001$),

suggesting that BEP supplementation led to a more modular and compartmentalized network over time.

Figure 4. Co-occurrence network analysis of maternal and infant microbiome across time points by groups

(A, B) Co-occurrence networks for (A) maternal samples at four time points and (B) infant samples at two time points. Nodes represent microbial taxa, with edges indicating significant co-occurrence relationships between taxa. Node size is proportional to the degree (number of connections), and colors indicate different communities identified by Louvain clustering.

(C) Topological properties of the ecological networks depicting co-occurrence of the gut microbiome, including nodes, edges, clustering coefficient, network density, network diameter, Louvain community count and modularity for both maternal and infant samples. Significant differences

between the intervention and control groups were determined using a permutation test (1,000 permutations), with significance set at $p < 0.05$.

4. The statistical analysis in general has been quite ordinary, especially for a journal of this caliber. There are probably too many issues to mention, but I will summarize the three key ones here. First, ANCOM-BC was used for differential abundance analysis, a method that has consistently exhibited a high empirical FDR in recent benchmarking studies (PMIDs: 35421994, 38701410). Moreover, a recent highly cited publication highlighted the inconsistency of current differential abundance methods (PMID: 35039521), indicating that reliance on only one method is risky. It should not be too difficult to run multiple methods to identify the most consistent signals in the data.

Reply: Thank you for your detailed feedback on the statistical analysis, which we greatly appreciate. First, regarding the use of ANCOM-BC for differential abundance analysis, we want to clarify that we employed ANCOM-BC2, which was recently published in *Nature Methods* and represents an advancement over the original ANCOM-BC. This method includes sensitivity analysis for pseudo-counts, making it particularly robust for our dataset. While we acknowledge concerns raised in recent benchmarking studies regarding the empirical FDR associated with certain methods (PMIDs: 35421994, 38701410), it is important to note that ANCOM-II and ALDEx2, as highlighted by PMID 35039521, have consistently produced the most reliable results across studies and have shown strong agreement with the intersect of results from various approaches.

To address the potential risks of relying on a single method, prior to submission, we have already conducted a comprehensive analysis using multiple differential abundance methods, including ALDEx2, TreeclimbR, MaAsLin2 and ANCOM-BC, with various normalization techniques. While MaAsLin2 did not return any significant results, among the other three methods, ANCOM-BC provided the most robust and consistent results for our dataset. Thus, we opted to use the updated version ANCOM-BC2 here. The outcomes of these comparative analyses are included as supplementary material to this rebuttal letter for transparency and can be added to the main paper should the reviewer and editor find this informative.

Regarding the concern about the high FDR, we would like to emphasize that our study reported relatively few significant differences, which suggests that FDR inflation is not a major issue in our findings. Instead, our primary concern has been ensuring adequate statistical power, particularly given the limited number of significant results across time points. We believe that the results we report are both reliable and conservative.

Lastly, while our associative statistical methods might be considered more ordinary, we have also employed methods from causal inference to strengthen our mechanistic claims. This approach, which is not commonly seen in microbiome research, provides an additional layer of rigor to our study.

5. Second, the follow-up analysis of the differential abundance analysis did not include an enrichment analysis, which was surprising as it would have been more appropriate to investigate the microbial functional pathways based on the differential abundance analysis results.

Reply: Thank you for your insightful suggestion regarding the inclusion of an enrichment analysis following the differential abundance analysis. We fully agree that investigating microbial functional pathways based on differential abundance results can provide valuable insights. Based on this point of feedback, we did perform a taxon set enrichment analysis to supplement our findings of the section ‘BEP supplementation is associated with maternal microbial abundances’ (line 927), but this did not yield any significant results. This outcome was likely due to the limited number of differential species identified in our differential abundance analysis, which constrained the statistical power required to detect meaningful enrichment in microbial functional pathways.

Given these limitations, we chose not to include the enrichment analysis in the current manuscript to avoid presenting potentially inconclusive or non-informative results. Nonetheless, we acknowledge the potential of this approach.

6. Finally, the mediation analysis was surprisingly limited to microbial diversity alone, despite the availability of multiple tools for conducting per-feature mediation analysis (individual microbial features as mediators). Such an analysis might provide more comprehensive and persuasive evidence to support the established causality.

Reply: We agree that incorporating per-feature mediation analysis could provide a more detailed understanding of how specific microbial features might mediate the effects of intervention. Though we would like to stress that using microbial features as multivariate mediators is not a straightforward endeavor, and still an active area of research, as the causal relationships between these features are unknown, which opens up paths of mediator-outcome confounding. Such intermediate confounding violates one of the core assumptions of causal mediation analysis, and hence makes it impossible under the current state-of-the-art to estimate natural (in)direct effects. Nonetheless, we did still strive to expand our mediation analyses by taking into account this possible source of intermediate confounding and using it to make estimates of interventional (in)direct effect. These interventional effects quantify the impact of the exposure on the outcome when the mediator (and potentially other intermediate confounders) is intervened upon (i.e., set to specific values) rather than left to vary naturally. In this way, they are similar to controlled (in)direct effects, except that the mediator is set to the value it would attain under a specific intervention on the exposure, rather than being fixed at an arbitrary level. If all relevant confounders, including intermediate ones, are accounted for, these interventional effects can provide valuable insights even when the cross-world assumption does not

strictly hold. To gain insight into the causal role of the microbiome in BEP's effect on the various health outcomes discussed in this manuscript, we estimated these interventional (in)direct effects using individual taxa as mediators. We furthermore employed a similar approach to estimate interventional effects using genes as mediators. We have updated the manuscript to reflect these comprehensive analyses.

Line 917 (track changes mode):

Microbiome modulation mediates birth outcomes and infant growth in response to BEP supplementation

We conducted causal mediation analyses to elucidate the role of the microbiome (through changes in either Shannon diversity or individual taxon abundance) in the downstream effects of BEP supplementation on birth outcomes (i.e., SGA, gestational age, birth weight and length) and infant growth metrics (i.e., LAZ, WAZ and WLZ) at 3 and 6 months of age.

The mediation analyses using the Shannon diversity of the microbial communities as a mediator revealed significant natural direct effects of the intervention on gestational age of the infants at birth (0.56 weeks; 95% CI: 0.05 – 1.08 weeks), as well as on WAZ at 3 months (0.43; 95% CI: 0.03 – 0.83) and at 6 months (0.43; 95% CI: 0.04 – 0.82) and on the odds of stunting at 6 months (-11.6%; 95% CI: -21.5 – -1.6%) (Figure 8A). However, no significant natural indirect effects were observed, indicating that these effects were not mediated by shifts in microbial diversity. Therefore, we conducted a multivariate mediation analysis using individual taxon abundances.

In the second trimester, specific taxa exhibited a significant mediating effect on birth outcomes. The abundances observed after the intervention of species from order Oscillospirales (UMGS911 sp900545935), *Lachnospira* (sp900316325), *Faecalibacterium prausnitzii*, and *Butyricicoccus A* (sp002395695) had significantly positive effects on gestational age, birth weight, and length. Conversely, the intervened abundances of the order Bacteroidales (RC9 sp900546925 and RC9 sp000434935), *Oscillospiraceae* (CAG-83 sp900547745), *Phascolarctobacterium succinatutens*, *Prevotella* sp900553155, *Gastranaerophilaceae* (UMGS1585), and *Prevotellamassilia* sp900543155 had negative effects on anthropometric outcomes (Figure 8B).

In the third trimester, *Ruminococcus bicirculans* and species from the order Christensenellales (CAG-1435), Bacteroidales (RC9 sp900546925), Oscillospirales (UMGS1002), and *Lachnospiraceae* (CAG-632 sp000431515) were associated with decreases in gestational age, birth weight, and length in mothers receiving BEP supplementation. Meanwhile, *Oscillospiraceae* CAG-110 sp900556935 showed a positive mediating effect gestational age, birth weight, and length in mothers receiving BEP supplementation, without affecting the odds of SGA (Figure 8C).

At 1 to 2 months of age, the abundances of *Bifidobacterium breve* and *Streptococcus* sp000187445 had significant mediating effects on both WAZ [*B. breve*: -0.13 (-0.24 – -0.01); *S. sp000187445*: 0.07 (0.01 – 0.12)] and WLZ [*B. breve*: -0.15 (-0.28 – -0.02); *S. sp000187445*: 0.08 (0.02 – 0.14)] at six

months (Figure 8D). *Bifidobacterium infantis* also displayed significant mediating effects on LAZ and WAZ at 3 and 6 months, though with negligible effect size (Table S3).

We also conducted causal mediation analysis on gene-level data by selecting the top 500 genes. While specific genes exhibited significant mediating effects on birth outcomes and infant growth, subsequent gene set enrichment analysis did not reveal distinct, significant microbial pathways. The complete results from the causal mediation analyses are available in Table S3.

Figure 8: Mediating effect of the microbiome on birth outcomes and infant anthropometry.

(A) Estimates for the Natural Indirect Effect (NIE) and the Natural Direct Effect (NDE) of the microbiome diversity on gestational age at birth and on length for age at six months. Results are

superimposed on the causal directed acyclic graphs used in these analyses. The model with maternal Shannon diversity as mediator included maternal stratum data as covariates (level of education, marital status, language, religion, ethnicity, number of jobs, socio-economic status, dietary diversity score, and health center), as well as maternal anthropometry taken at inclusion into the trial (age, height, BMI and hemoglobin level). The model with infant microbiome diversity as a mediator also included the age at which the infant stopped exclusively breastfeeding as a covariate. Infant microbiome diversities were measured at 1 to 2 months of age. **(B)** Estimates for the interventional direct and indirect effects of each individual species' abundance in the maternal microbiome in the second trimester, **(C)** the third trimester, and **(D)** in the infant microbiome at 1-2 months of age. The complete results of these mediation analyses are available in Table S3.

7. A minor comment to conclude my comments on the statistical methods: the treatment effect of the randomized intervention was estimated using the doubly robust TMLE. What are the components of the multiple SuperLearner models included in this exercise? How sensitive are the results concerning various combinations of base models? Given that there are so many parameters involved, I was surprised to see no comprehensive sensitivity analysis of hyperparameters is conducted which is a major drawback of this particular analysis.

Reply: Thank you for your insightful comments regarding the use of the SuperLearner model in our TMLE analysis. We included the following algorithms in the SuperLearner model: SL.glmnet, SL.ranger, SL.earth, SL.glm, SL.randomForest, SL.step, and SL.glm.interaction. We have now added this information to the Methods section for clarity.

The SuperLearner algorithm is designed to optimize and combine these models, thereby enhancing robustness and reducing sensitivity to any particular set of hyperparameters. The ensemble approach of SuperLearner inherently performs a form of sensitivity analysis by selecting and weighting the best-performing models from the ensemble. This process ensures that the final model is the most robust possible combination of the included algorithms.

Furthermore, the use of SuperLearner for sensitivity analysis has been supported in the literature, as evidenced by the study referenced (PMID: 36819084), which demonstrates its application for this purpose. Given these considerations, we believe that our approach effectively addresses concerns related to hyperparameter sensitivity.

Editor:

In summary, despite collecting a rich dataset across time points, treatment groups, and experimental units, the analyses fall short in many ways in generating solid, convincing biological interpretations. Unfortunately, it would take quite a bit of work to complete the study. Without a major comprehensive re-analysis, it is not clear if this manuscript serves as a meaningful contribution to justify publication in Nature Communications.

R
e
p
l
y
:

We have conducted an extended analysis to strengthen our findings and enhance the biological interpretations. We have summarized the key changes and improvements in our cover letter for your review. We believe these revisions significantly contribute to the manuscript's quality and clarity, justifying its potential for publication in Nature Communications.

Reviewer #1 (Remarks to the Author):

I thank the authors for responding to most of the comments suitably. As the manuscript is considerably altered following the first version, I have further comments that need to be addressed below. In general, I am not sure that some of the additional figures/analyses are necessary outside of what was proposed in the original reviewers comments. Simple, clear figures showing the most important results should be saved for the main text. Anything else should be saved for supplementary. I appreciate that there are many detailed suggestions, however these are all suggested to help clarify these interesting results. I provide detailed points below:

1. Although the larger trial protocol has been published previously, it is critical to detail when the intervention was started and any other critical details. Please provide details and statistics for average week of gestation at start of intervention for each group
2. Related to this point above, please provide a table that details the baseline characteristics of the study population, split by intervention groups and with statistics/p-values showing differences between intervention groups. Ideally this table will also show details of these characteristics for the larger MISAME-III population to show whether this subset of participants is representative of the larger trial population. A simple table was present in the initial manuscript but not in the revised manuscript. Please re-include this and add details requested above.

Reply to 1 & 2:

Thank you for your suggestions. We have re-included a table summarizing the baseline characteristics of the BioSpé subset population alongside data for the larger MISAME-III trial population. This table includes statistical analyses and p-values to highlight any differences between the intervention groups, as well as details that demonstrate the representativeness of the BioSpé subset relative to the larger trial cohort. Additionally, we have incorporated the details on the average gestational week the intervention began for each group. This information is now clearly stated in the text and Table S1.

Added to line 80:

“The mean (SD) gestational age at birth was 39.8 (1.72) weeks, and the gestational age at which the intervention began was 9.85 (3.58) weeks, with no significant differences observed between the intervention and control groups.”

Table S1. Baseline characteristics of MISAME-III participants and the subset participants

Characteristics	MISAME-III					Subset		
	Pre- and Postnatal Intervention (n = 475)	Prenatal Intervention (n = 462)	Postnatal Intervention (n = 471)	Control (n = 489)	P value	Pre-and Postnatal Intervention (n = 71)	Control (n = 81)	P value
Health center catchment area, %					0.99			0.99
Boni	23.6	22.1	22.1	23.5		15.5	13.4	
Dohoun	11.8	9.50	9.80	11.0		12.7	11.0	
Dougoumato II	18.5	17.5	18.3	18.6		22.5	15.9	
Karaba	9.9	10.6	10.2	10.0		12.7	12.2	
Kari	17.1	19.3	20.0	17.6		21.1	23.2	
Koumbia	19.2	21.0	19.7	19.2		15.5	24.4	
Household level								
Household food insecurity*, %	67.4	71.0	67.9	64.2	0.61	67.6	70.7	0.98
Wealth index, 0 to 10 points	4.62 ± 1.76	4.69 ± 1.73	4.55 ± 1.77	4.52 ± 1.77	0.45	4.70 ± 1.67	4.64 ± 1.87	0.17
Maternal								
Age, years	24.8 ± 6.24	25.1 ± 6.07	25.0 ± 6.33	25.1 ± 6.03	0.83	23.4 ± 5.56	25.2 ± 5.51	0.20
Ethnic group, %					0.31			0.37
Bwaba	55.6	59.1	57.5	56.9		49.3	53.7	
Mossi	35.2	33.8	35.0	35.6		39.4	32.9	
Others	9.26	7.14	7.22	7.57		11.3	13.4	
Religion of pregnant woman, %					0.44			0.91
Animist	20.8	23.8	23.6	22.9		22.5	22.0	
Muslim	42.9	41.6	42.0	42.3		47.9	43.9	
Catholic	12.4	13.6	15.1	13.9		7.04	6.10	
Protestant	21.3	16.9	15.7	16.8		19.7	25.6	
No religion, no animist	2.53	4.11	3.40	4.09		2.82	2.44	
Primary education and above, %	41.1	42.4	44.6	39.5	0.54	38.0	45.1	0.36
Number of jobs, %					0.68			0.33
0	61.7	60.0	61.6	59.7		64.8	53.7	
1	33.1	36.8	35.5	36.2		33.8	42.7	
2 and above	5.26	3.25	2.97	4.09		1.41	3.66	
Parity, %					0.18			0.41
0	24.2	22.5	23.8	20.0		29.6	20.7	
1-2	35.4	30.3	35.9	36.2		33.8	39.0	
3 or more	40.4	47.2	40.3	43.8		36.6	40.2	
Weight, kg	58.7 ± 8.88	58.0 ± 8.44	57.6 ± 8.24	58.4 ± 9.07	0.29	57.7 ± 8.30	60.2 ± 12.2	0.11
Height, cm	163 ± 5.75	163 ± 6.30	163 ± 5.88	162 ± 6.16	0.36	162 ± 4.96	163 ± 6.72	0.63
Body mass index, kg/m ²	22.1 ± 2.99	21.9 ± 2.73	21.8 ± 2.73	22.2 ± 3.00	0.13	22.0 ± 3.16	22.6 ± 3.74	0.16
Hemoglobin, g/dL	11.3 ± 1.48	11.2 ± 1.56	11.4 ± 1.53	11.4 ± 1.46	0.28	11.9 ± 1.38	11.6 ± 1.46	0.10
Anemia, hemoglobin <11 g/dL, %	38.5	41.6	38.0	36.8	0.49	26.8	34.1	0.06
Gestational age at birth, weeks	39.9 ± 1.71	40.1 ± 1.33	39.9 ± 1.63	39.7 ± 2.16	0.01	40.0 ± 1.63	39.6 ± 1.77	0.45
Gestational age at inclusion, weeks	11.6 ± 4.45	12.2 ± 4.50	11.8 ± 4.38	11.8 ± 4.61	0.24	9.61 ± 3.23	9.97 ± 3.73	0.79
Intervention length when Sample collection, weeks								
Mother – Tri2						13.3 ± 2.31	13.9 ± 2.43	0.51
Mother – Tri3						22.0 ± 3.49	21.5 ± 4.02	0.72
Mother – Pn12						39.3 ± 3.52	38.3 ± 4.56	0.46
Mother – Pn56						56.3 ± 3.35	55.6 ± 4.44	0.74
Infant – Pn12						39.5 ± 3.45	38.4 ± 4.38	0.40
Infant – Pn56						56.3 ± 3.42	55.7 ± 4.59	0.78

Values are percentages or means ± standard deviation s.

Group comparisons were conducted using ANOVA for continuous variables and chi-squared tests for categorical variables.

*Assessed using FANTA/USAID's Household Food Insecurity Access Scale.

3. Also related to point 1 above, if there is large variation between participants in the length of time between intervention initiation and fecal sample collection, it will be important to adjust for that. Can the authors clarify this point and adjust for this if necessary?

Reply:

As shown in Table S1, there was no significant difference between participants in the length of time between intervention initiation and fecal sample collection. Therefore, it was not necessary to adjust for this variable in our analyses.

4. The top left panel in Fig 1 is only suitable if there are no significant differences in any of these factors between groups, because this image averages the entire population. If there are any differences between groups, I would be inclined to present all of this data in the baseline characteristics table.

Reply:

Since there are no significant differences in any of these factors between groups, we have decided to retain the top left panel in Figure 1 and include the baseline characteristics table as supplementary material. Additionally, we would like to clarify that this image does not average the entire population; rather, the first two rows of charts represent the intervention group, while the remaining rows correspond to the control group.

5. I am still not entirely happy with the details for how the participants for this sub-study were selected and the cited protocol for BioSpé doesn't appear to elaborate. I understand that this may be a convenience subset (usually limited by funding etc), however it would be important to state if so, and most importantly whether these participants were selected at random from the larger trial. This maybe just needs a little clarification, but as stated above, it would be extremely helpful to provide stats to show whether this sub-population were representative of the larger trial population. Otherwise, this could be a biased sub-population

Reply:

Thank you for raising this important concern. The participants in this sub-study were not randomly selected from the larger trial. The launch of the BioSpé study was significantly delayed due to pandemic-related disruptions in the shipment of scientific sampling materials to Burkina Faso. As a result, we recruited the 300 participants with the earliest gestational ages at study launch to maximize the duration of prenatal follow-up despite these delays. Additionally, we ensured a balanced distribution between the intervention and control groups to maintain comparability.

Specifically, we selected the 75 women with the earliest gestational ages within each trial arm. This approach was designed to maintain a minimum sample size of 60 participants per group at each time point. Second trimester sample collection was limited because, by the time recruitment began, many women were already in their third trimester. This selection strategy also accounted for an anticipated 20% dropout rate, balancing feasibility with the need for sufficient statistical power for longitudinal analysis.

As shown in Table S1, the baseline characteristics of this subset population are comparable to those of the full MISAME-III cohort. The only difference is a slightly lower prevalence of anemia in the intervention group within the subset; however, this difference is not statistically significant. We have clarified this information in the main text to ensure transparency regarding the representativeness of the subset population.

Added to line 55:

Participants for this sub-study were selected based on gestational age to maximize second-trimester recruitment, ensuring a minimum sample size of 60 at most time points. Accounting for a 20% dropout rate, we aimed to select 75 participants per group, balancing feasibility with achieving sufficient statistical power for longitudinal analysis.

Added to line 68:

The baseline characteristics of the subset population were comparable to the full MISAME-III cohort, except for a slightly lower percentage of anemia in the intervention group within the selected subset, that is not statistically significant. (Table S1).

6. Line 79-80, please clarify if “higher” WAZ and WLZ were significant or not.

Reply:

Thank you for pointing this out. We have clarified this by adding a sentence to indicate that the observed differences were not statistically significant.

Correction (line 86):

“This difference was also reflected in higher Weight-for-Age Z-scores (WAZ) and Weight-for-Length Z-scores (WLZ) during the first six months of life in the intervention group. However, these differences were not statistically significant.”

7. Line 104 doesn’t match the figure 2c that it refers to. It states that prenatal and postnatal Shannon diversity are 4.5 and 3.5 respectively but fig 2c appears to show that median values are all ~4.5 for every group and timepoint. Can you clarify these statistics?

Reply:

Thank you for highlighting this discrepancy. The postnatal samples referenced in the original analysis included both maternal and infant samples, which was incorrect. Upon reanalysis, filtering out the infant samples, we found that the median postnatal maternal Shannon diversity should be 4.10 bits, not 3.52 bits. We have corrected this error and revised the manuscript accordingly.

Correction (line 111):

“Shannon diversity was higher in prenatal samples compared to postnatal samples, but the difference was not statistically significant ($p = 0.08$, Wilcoxon rank sum test), with median values of 4.53 bits and 4.10 bits, respectively (Figure 2C).”

8. The results for increasing Shannon diversity by BEP appear extremely subtle. As suggested above, did you (or could you) adjust for days/weeks since start of the intervention? For example it is still not clear whether the Tri2 samples are taken days or weeks after the start of the intervention, so it is difficult to interpret this significant result.

Reply:

Thank you for highlighting this point. After carefully re-checking, our data, we confirmed that the second trimester (Tri2) fecal samples were collected approximately 13 weeks after the start of BEP supplementation, as indicated in Table S1. While the results for Shannon diversity were indeed subtle, the statistical significance was robust after adjusting for potential confounders such as maternal baseline characteristics (e.g., age, BMI, hemoglobin levels, dietary diversity). Given the consistency in the timing of sample collection across participants, we did not find substantial variation in the time elapsed since the start of the intervention to warrant additional adjustment.

9. Related to this point above, can you please discuss somewhere how to interpret this result that the intervention would only have an effect prenatally and not postnatally when the intervention is being continued?

Reply:

The observed effect of BEP supplementation on Shannon diversity being significant during the prenatal period but not postnatally could reflect distinct physiological and microbiome dynamics between these phases. Prenatally, the maternal gut microbiome undergoes substantial remodeling to adapt to the metabolic demands of pregnancy, such as nutrient allocation for fetal growth. BEP supplementation may interact with this naturally occurring remodeling process, amplifying diversity changes during this critical period. Postnatally, the maternal microbiome stabilizes and shifts towards a composition more typical of non-pregnant states. This stabilization may reduce the magnitude of microbiome changes detectable due to supplementation. We have added this discussion to the manuscript to address this observation.

Added to Discussion (line 296):

“The effect of BEP supplementation on Shannon diversity was significant during the prenatal period but not postnatally. This may reflect the distinct physiological and microbiome dynamics between these phases. During pregnancy, the maternal gut microbiome undergoes substantial remodeling to meet the metabolic demands of fetal growth³⁰, and dietary supplementation may interact with these changes, amplifying microbial diversity. Postnatally, the maternal microbiome transitions toward a more stable state over time as the body adjusts to lactation and recovery³¹, potentially reducing the detectable impact of supplementation.”

10. The distribution of data points in the the PCoA in Fig 2F seem highly unusual (and in the infant panel it is skewed by 3 data outlying data points). I would imagine that this is due to some issue with transformation/filtering/normalization of gene count data. I would suggest to re-check this data before final publication.

Reply:

Thank you for your observation regarding the distribution of data points in the PCoA plots. We have carefully rechecked our data processing pipeline and visualization methodology. Initially, the raw gene counts were used for calculating Bray-Curtis distances, which could have introduced skewness in the results due to the compositional nature of microbiome data.

To address this, we updated our preprocessing steps by normalizing the gene counts using Total Sum Scaling (TSS), followed by a log transformation ($\log_1 p$) to reduce the effects of high-abundance genes and stabilize variance. These transformations account for compositionality and zero-inflation, ensuring that the data is appropriately scaled for Bray-Curtis dissimilarity calculations.

Additionally, using the transformed data, PERMANOVA analysis revealed that temporal variation significantly shapes microbial gene beta-diversity, with significant differences observed over time in both maternal ($p = 0.003$) and infant ($p = 0.002$) samples. However, no significant differences were attributed to the intervention in either maternal ($p = 0.155$) or infant ($p = 0.704$) samples. These findings have been updated in the main text for clarity.

The revised PCoA plots now provide a more accurate and interpretable representation of microbial community composition, as shown in the updated figures. We also verified that the outlying points identified in the infant panel are consistent with the dataset and not due to technical artifacts. These points represent samples with low overall gene richness but are biologically valid and align with the diversity trends observed in infant gut microbiomes.

We appreciate your suggestion, as this adjustment improves the robustness of our results and addresses potential biases in the earlier analysis.

Correction (line 131):

“Although beta-diversity of gene data showed diversification of the infant metagenome over time ($p = 0.003$), no significant differences were attributed to the intervention ($p = 0.155$). Similarly, in infant samples, the metagenome showed significant diversification over time ($p = 0.002$), but no significant differences could be attributed to the intervention ($p = 0.704$) (Figure 2F). These results indicate that temporal changes in the gut microbiome play a more prominent role in shaping microbial gene diversity than the intervention itself.”

Figure 2. Relative abundance, diversity, and gene richness in maternal and infant samples across time points by groups.

11. As above, it will be important to attempt to interpret why gene count data in 2E is only significant at 1 postnatal time point

Reply:

We have added the following paragraph in the Discussion section.

Added to Discussion (line 302):

“The significant effect of BEP supplementation on gene richness at 1-2 months postpartum may reflect the dynamic shifts during early lactation, as maternal metabolism adapts to meet increased energy demands. These changes, interacting with BEP supplementation, could enhance microbial functional potential, explaining the observed increase in gene richness. As lactation stabilizes, the effect of supplementation diminishes, reducing differences at later stage.”

12. I am not convinced that Fig 3B is very informative. I would suggest to simplify this to simply show total no. shared species between mothers-infants in each treatment group. Given the depth of this sequencing as illustrated in Fig. S1, it is disappointing not to see strain-sharing data, however I appreciate that this is a larger undertaking.

Reply:

Thank you for your comment. Upon re-checking our analysis, we confirm that these data represent shared strains, as the matching between mothers and infants was based on genomes. We have revised

t
h
e
r
e
e
We would like to retain Figure 3B because it provides valuable insights into the number of shared strains between mothers and infants at each time point, showcasing temporal trends and group-specific differences. While the total number of shared strains is already stated in the main text (line 154), this figure adds granularity by illustrating how these shared strains are distributed over time and across groups. Such temporal insights are crucial for understanding the dynamics of mother-infant microbial sharing.

v

a
n
t
13. Fig 3C needs error bars on each data point. Please also indicate more clearly in the figure that the $p=0.02$ is from a comparison of the first 2 maternal time points

p
a
t
h
14. Furthermore, I think Fig 3C needs to be adapted to either separate out maternal and infant samples into separate panels, divide the y axis or log transform values (probably the easiest option) so that maternal value can be seen more clearly as the abundances of all maternal values are so low, it is impossible to visualize increases/decreases

o

Reply to 13 & 14:

We appreciate your thoughtful suggestions. We have added error bars to each data point in Figure 3C and applied log transformation to address the differences in scale between maternal and infant

h

e

m

a

i

abundances, ensuring better visualization of maternal values. Additionally, the placement of the $p = 0.02$ has been adjusted to be closer to the first two maternal time points in the figure. Due to limited space, further adjustments to the label placement may not significantly improve clarity. We have explicitly clarified in the main text that the p -value refers to the comparison between the first two maternal time points.

15. It is great to see some longitudinal analysis in Fig 3D. However, I am not entirely clear on the stats methods (albeit I am not a statistician). It is still most suitable to assess statistical differences between intervention groups, while accounting for changes over time. It seems to me that the statistical differences have only been assessed over time within treatment groups? For example, I am surprised to see that the *B. fragilis* enrichment noted in the first version of this manuscript has not been sustained in this new analysis. It appears to me from Fig 3D that this is a relatively strong effect of the intervention (in addition to some prevotella species). I would suggest to revisit these statistics to confirm that you are capturing the significant differences between intervention groups, whilst adjusting for the longitudinal effects.

Reply:

Thank you for your insightful observations. To clarify, the purpose of Figure 3D is to capture longitudinal changes in abundance over time within each treatment group and to visualize how these changes differ between groups. The statistical differences between groups at each time points were analyzed using differential abundance analysis method (ANCOM-BC2 here) and reported in Figure 5, where we specifically highlight taxa with significant group-level difference. For example, *B. fragilis* remains a significant result in Figure 5 (third panel) and was discussed in the main text (line 201).

Regarding the changes made to Figure 5, the new version visualizes all taxa with significant group-level differences where the effect size is greater than 0.5. This is a broader threshold compared to the older version, which only included taxa with an effect size (\log_2 fold change) greater than 1. By using this updated threshold, we provide a more comprehensive view of the taxa impacted by the intervention. For Figure 3D, the longitudinal visualization complements Figure 5 by showing temporal trends. Using mixed-effects model, the analyses have accounted for repeated measures within individuals and adjusted for longitudinal effects.

We hope this clarification resolves the concerns regarding the statistical approach and interpretation of the results.

16. Results in Fig 3E are interesting but need a little clarification. Firstly, is this analysis not a form of analysis conducted in Fig 2D and therefore could they not be presented together? Secondly, it is not clear in the mother data which time points are being compared for when analyzing Unifrac distances (is it all timepoints or first and last?).

Reply:

To clarify, the analysis presented in Figure 3E is distinct from that in Figure 2D, despite both using UniFrac distance. Figure 2D focuses on the dissimilarity of microbiome data at each time point, comprising microbial composition between treatment groups at a single time point. In contrast, Figure 3E examines temporal changes in microbiome composition (assessed using intra-subject distances) across consecutive time points. In the analysis shown in Figure 3E, a mixed-effect linear regression

model was used, where intra-subject distance serves as the dependent variable and the time interval (in days) between consecutive time points serves as the independent variable. Subject ID is included as a random effect to account for repeated measures. The x-axis represents the beta coefficient estimates from each mixed-effects model, indicating the rate of temporal change. Therefore, the analysis incorporates all time points.

We have revised the relevant part of the main text (line 170) to improve clarity:

“..., as evidenced by mixed-effect linear regression models with intra-subject distance (calculated using weighted UniFrac) as the dependent variable and days interval between consecutive time points as the independent variable (Figure 3E).”

17. The right panel on Fig 2F is also quite confusing. It seems as if the horizontal lines are error bars? If so, please make them smaller like typical error bars?

Reply:

Thank you for pointing this out. We assume you are referring to Figure 3F, not Figure 2F. Figure 3F visualizes changes in UniFrac distances between consecutive time points, highlighting the dissimilarity in microbial composition over time. The horizontal lines are indeed error bars, and we have reduced their size to conform to typical error bar conventions for improved clarity and

r

e

Add to line 173:

Similarly, Figure 3F revealed distinct temporal patterns in microbiome composition. In maternal samples, microbiome dissimilarity increased progressively from the second to the third trimester, peaked between late pregnancy and early postpartum, and then stabilized. The shift was more pronounced in the intervention group, but it was not statistically significant. In infants, microbiome composition exhibited greater shifts over time compared to maternal samples, with slightly higher UniFrac distances observed in the intervention group, though this difference was not statistically significant.”

y

.

W

e

h

o

p

e

t

h

e

r

e

v

i

s

e

d

Figure 3. Temporal stability of maternal and infant gut microbiomes across time points by groups.

18. Network analysis portrayed in Fig 4 adds another element to the analyses but without interpretation it is not that meaningful. Is it possible to analyse which taxa are important or changing in these networks between intervention groups and over time. For example, are any of the taxa from the earlier analyses (e.g. *B. fragilis* or *Prevotella*) important nodes in these networks? Without concrete interpretation of these networks and what they mean with regards to the intervention (i.e. what does increased modularity really mean – is it good/bad?), this analysis may be better suited in supplementary or removed.

Reply:

Thank you for your thoughtful comment. Network analysis provides a comprehensive overview of how all taxa interact within the microbial community, allowing us to explore the potential impact of the intervention on the structure and dynamics of these interactions. While it is possible to examine specific taxa from earlier analyses (e.g., *B. fragilis* or *Prevotella*) within these networks, our primary

goal in conducting this analysis was to treat the microbial community as a whole to assess how the intervention influences overall community structure, rather than focusing on individual taxa. We acknowledge that this intent may not have been clear previously, and we have clarified this in the revised text. The interaction of specific taxa with others may not necessarily be impacted by the intervention, which is why we did not annotate or highlight specific taxa in the network plots. Instead, we aimed to capture changes in broader network properties, such as modularity, density, and community structure, which provide insight into the global effects of the intervention. In this vein, it is more similar to the alpha- and beta-diversity analyses than it is to the taxon-based DAA or mediation analyses.

To address your concern about the interpretation of these networks, we have added detailed explanations to elaborate on the significance of the observed changes. Specifically, increased modularity, as seen postnatally in the intervention group, likely reflects a shift toward a more specialized network structure, where microbial taxa form distinct and cohesive clusters with limited interaction between groups. This compartmentalization can enhance the resilience of the microbiome to environmental changes, potentially safeguarding key functional groups from external disturbances. Such structural changes may support gut homeostasis and promote a more stable and less inflammatory postnatal environment, aligning with our findings of reduced pro-inflammatory pathways and increased abundance of *B. fragilis*.

We believe that retaining Figure 4 in the main text is important as it adds another layer of observation regarding the intervention's impact on the microbiome, complementing the other analyses. We have revised the Discussion section to incorporate these interpretations and believe this enhances the relevance and clarity of the network analysis.

Added to the Discussion (line 333):

“Increased modularity may reflect a shift toward a more specialized network structure, where microbial taxa form distinct and cohesive clusters with limited interaction between groups. This compartmentalization can enhance the resilience of the microbiome to environmental changes, potentially safeguarding key functional groups from external disturbance³⁸.”

19. Overall there are many figures. I would reserve key ones for the main text. I think Fig 7 could go in supplementary. Arguably fig 4 could also go in supplementary, as although the networks looks nice, the figures alone don't tell you much about the results

Reply:

We agree that Figure 7 could be moved to the supplementary materials, and we have done so. However, we would like to retain Figure 4 in the main text, as it provides a critical layer of analysis regarding the intervention's impact on the microbial community structure.

20. In text describing mediation analysis (lines 256-261) please indicate direction of mediating effects rather than just “significant mediating effect”.

Reply:

The symbols (- or not) indeed indicate the direction of the mediating effect, but we understand that this may not have been sufficiently clear. We have revised the text to explicitly describe the direction of these effects for better clarity.

Revised text (line 272):

“At 1 to 2 months of age, the abundances of *Bifidobacterium breve* showed a significant negative mediating effect on both WAZ (-0.13; 95% CI: -0.24 – -0.01) and WLZ (-0.15; 95% CI: -0.28 – -0.02) at six months, indicating that higher levels of *B. breve* were associated with lower WAZ and WLZ scores. Conversely, *Streptococcus* sp000187445 exhibited a significant positive mediating effect on WAZ (0.07; 95% CI: 0.01 – 0.12) and WLZ (0.08; 95% CI: 0.02 – 0.14), suggesting that its abundance was associated with higher WAZ and WLZ scores (Figure 7D).”

21. The mediation analysis results presented in Fig 8B-D is very interesting but difficult to comprehend. As far as I can tell, the blue lines are the direct effect of the intervention on each outcome (gestational age, birth weight etc). The yellow lines are the mediating effect of each taxon. However, if this is correct, then the blue lines should all be exactly the same within each panel? The other alternative is that the blue lines are the direct effect of each taxon on the birth/growth outcome, independently of the intervention. If this alternative is correct, then the results are very difficult to interpret, as almost all results show opposing results for NDE and NIE. For example, in Fig 8C left panel (gestational age), *Ruminococcus D bicirculans* has a very strong positive direct effect on gestational age, increasing it by 5 weeks. However, it has a strong negative mediating effect of the intervention on gestational age, reducing it by ~2.5 weeks. Apart from these effect sizes seeming absolutely huge (are you sure these aren't days?), the results are difficult to comprehend and interpret. I would advise re-checking this analysis and if it remains the same, discussing how to interpret these findings in the discussion.

Reply:

The mediation analyses presented in Figure 8B-D (now Figure 7B-D) evaluate the interventional (in)direct effects, rather than the natural direct or indirect effects. This approach allows us to robustly estimate a causal mediating effect for each microbial taxon's abundance while addressing the interconnected causal linkages between the microbiota, a challenge referred to in the literature as intermediate confounding.

In this analysis, the yellow lines represent the estimated indirect effect of each taxon on the birth or growth outcome. Specifically, the indirect effect quantifies the difference in the outcome when the microbial taxon is at an abundance observed in the BEP-treated group versus when it is at an abundance observed in the untreated group (in essence, isolating the taxon's effect as triggered by the intervention, without compounding the direct effect of BEP into this estimate). Conversely, the blue lines represent the direct effect of BEP treatment on the outcome if the abundance of the microbial taxon (marked on the graph's vertical axis) were unchanged by the intervention. The variation in the direct effect estimates across taxa arises because the direct effect accounts for all other taxa besides the one indicated on the axis. This differentiation highlights the unique contribution of each taxon as a mediator. The estimate of direct effect on gestational age in Figure 8C (now 7C) hence does not indicate an effect of *Ruminococcus D bicirculans*, but rather the estimated effect that BEP would have had if *R. bicirculans*' abundance had not been affected by the intervention. We observe in this figure, that *R. bicirculans* counteracts this movement.

The observed opposing results for the direct and indirect effects for some taxa reflect the fact that we only included taxa with statistically significant indirect effects in the figure (i.e., indirect effect estimates whose confidence intervals do not cross zero). Taxa without meaningful indirect effects tended to show substantial overlap between the blue (direct) and yellow (indirect) lines. For a complete understanding, we have included the full set of results, including nonsignificant taxa, in the supplementary data (Table S4).

We appreciate your observation regarding the large effect sizes. These values are in weeks, not days, and have been rechecked for accuracy. As we realize that this type of analysis is not conventional in microbiome literature and not that straightforward in interpretation, we have added clarification in the discussion to aid in interpreting these findings.

Added to the Discussion (line 371):

“While we recognize that the counterfactual nature of indirect effects makes them difficult to interpret, these analyses highlight species that play a key regulating role in their ecosystem. Diverging indirect and direct effects, such as seen for *Ruminococcus D bicirculans* in the third trimester, indicate a regulating role, counteracting effects from the intervention and other microbiota. Conversely, indirect effects that outweigh or compound the direct effects highlight changes in birth outcome that are mechanistically caused by specific microbiota.”

Reviewer #3 (Remarks to the Author):

The authors have done a decent job revising the earlier manuscript. Most of the concerns raised in the first round of review have been resolved. In particular, the updated analyses as reported in Figures 3 and 4 represent some of the most interesting findings. Given the significance of the dataset generated, the paper is now in very good condition to be published.

Reply:

We sincerely appreciate your positive feedback and your acknowledgment of the revisions made to the manuscript. We are particularly pleased that you found the updated analyses in Figures 3 and 4 to be some of the most interesting findings. Your constructive comments during the earlier round of review greatly improved the clarity and depth of our work.